# Leveraging Self-Supervised and Supervised Embeddings for Memory-Efficient Experience-Replay Continual Learning

## Abstract

Catastrophic forgetting remains a key challenge in Continual Learning (CL). In replay-based CL with severe memory constraints, performance critically depends on the sample selection strategy - that is, which examples are stored for replay. Most existing approaches construct memory buffers using embeddings learned under supervised objectives. However, class-agnostic, self-supervised representations often encode rich, class-relevant semantics that are overlooked. We propose a new method, MERS- *Multiple Embedding Replay Selection*, which replaces the buffer selection module with a graph-based approach that integrates both supervised and self-supervised embeddings. Empirical results show consistent improvements over state-of-the-art selection strategies across a range of continual learning algorithms, with particularly strong gains in low-memory regimes. On CIFAR-100 and TinyImageNet, *MERS* outperforms single-embedding baselines without adding model parameters or increasing replay volume, making it a practical, drop-in enhancement for replay-based continual learning.

## 1 Introduction

*Continual Learning* (CL) focuses on acquiring knowledge from a continuous stream of data, where the distribution of information can change over time. Unlike traditional machine learning, which relies on static datasets and assumes that the data distribution remains fixed, many real-world scenarios involve environments that evolve over time, such as autonomous driving.

A core challenge in CL is *catastrophic forgetting* [18], the tendency of neural networks to lose previously acquired knowledge when trained on new tasks. Without mechanisms to preserve past information, models quickly forget earlier concepts, leading to degraded performance over time. This issue is especially pronounced in the *class-incremental learning* (CIL) setting, a particularly challenging variant of continual learning. In CIL, each task introduces entirely new classes, and the model must learn to classify all new and previously seen classes jointly.

Owing to this difficulty, *replay-based methods* have emerged as an effective approach to mitigate catastrophic forgetting in CIL. These methods maintain a small buffer of selected past examples and replay them during training on new tasks. However, when the buffer is small, as is often the case, the strategy for selecting the examples to retain becomes critical to overall performance. Most existing selection strategies rely on representations learned from supervised models, which may fail to capture the full diversity or structure of the data.

We propose a novel algorithm that unifies multiple representation spaces, each capturing distinct aspects of the data, within a graph-based selection framework (see Fig. 1). By leveraging nonparametric density estimation and localized coverage strategies, our method selects examples that offer better coverage and diversity across all embedding spaces. This multi-embedding approach can be integrated into existing selection algorithms (see review of related work below). It adapts their hyperparameters in a data-driven manner to the geometry of each embedding space.

We demonstrate that our method consistently outperforms single-embedding baselines across several continual learning scenarios and datasets. Notably, our improvements are most pronounced in low-buffer regimes, where efficient use of memory is crucial.

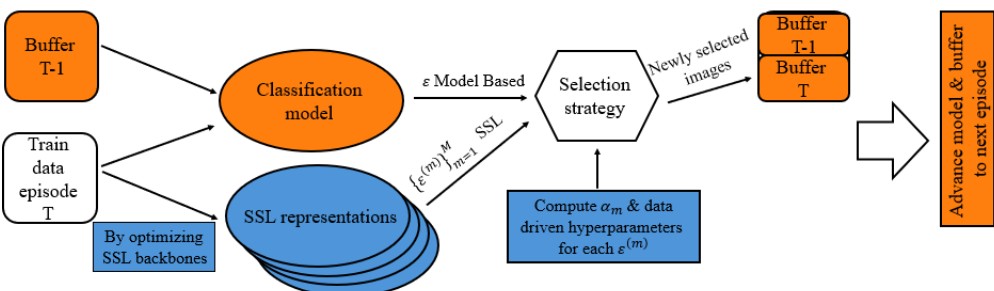

Figure 1: Illustration of our *MERS* in the class-incremental learning (CIL) setup: Training data is split into disjoint episodes, where at each episode only the current data is accessible and becomes unavailable afterward. Orange components denote the standard CIL process: the classification model and buffer are retained and advanced to the next episode. Blue components highlight our contribution, *MERS*, which augments CIL by training a self-supervised model from scratch at each episode, extracting its embeddings, and using them (together with the supervised model outputs) to guide buffer updates through a data-driven selection strategy.

## 2 RELATED WORK

**Continual learning paradigms**  CL approaches are often grouped into (i) regularization-based methods that constrain parameter updates to preserve prior knowledge (e.g., EWC [15], LwF [17]), (ii) architecture-based methods that expand capacity across tasks (e.g. HAT [22], DAN [28]) and (iii) Replay-based methods that maintain a small memory of exemplars for replay (e.g., ER [21], ER-ACE [7]). In CIL, rehearsal is particularly competitive under tight memory budgets because it is able to preserve decision boundaries as the label set grows [14].

**Selection strategies**  A central problem for Replay-based methods is *exemplar selection*. The iCaRL method employed a *Herding* algorithm to approximate each class centroid within a fixed feature space [14]. More recent strategies fall into two complementary families: (i) Gradient or conflict-oriented approaches, such as GSS [2], that prioritize samples whose loss gradients would be most altered by the impending parameter update, thereby directly mitigating catastrophic interference. (ii) Representativeness-oriented approaches, such as TEAL [23], that keep only the samples with the highest typicality, the inverse of their mean distance to the K nearest neighbors, ensuring that the buffer stores the most representative points.

**Coverage-based selection and its guarantees**  *ProbCover* formalizes tiny-buffer selection as covering a $k$-NN graph and offers a $(1 - 1/e)$ greedy guarantee [27]. *MaxHerding* smooths this objective with kernels, retains submodularity, and is robust to hyperparameters [3].

Prior Continual Learning heuristics, such as iCaRL's Herding-based exemplar selection [20], and Rainbow Memory's diversity through uncertainty strategy [5], compute coverage in a single embedding at a fixed scale, rendering them brittle to task heterogeneity. Our method introduces the notion of coverage to this line of work, expands coverage to multiple embeddings, and *adapts* locality per embedding using nonparametric statistics, which we find crucial in tiny-buffer regimes.

**Self-supervised representations for CL**  Self-supervised learning (SSL) captures class-agnostic invariances that naturally complement supervised features [24]. Contrastive methods such as SimCLR [11] and redundancy-reduction objectives like VICReg [6] yield rich embeddings without label supervision, while teacher-student paradigms like DINO [8] tighten view consistency.

These SSL representations have already demonstrated effective transfer to object detection, semantic segmentation, depth estimation, robotics manipulation, and few-shot recognition, often rivaling or surpassing supervised pretraining [24]. Yet most rehearsal-based CIL methods still choose exemplars solely in the *supervised* feature space of the current classifier, with only a handful operating purely in an SSL space, as mention in the Selection strategies part.

We instead *jointly exploit supervised and SSL embeddings*, ensuring that the memory preserves both class-discriminative and class-agnostic geometry. This dual-space strategy harnesses complementary signals and, as our experiments confirm, delivers consistent gains in the tiny-buffer continual-learning regime.

**Multi-view learning** This is an ML paradigm where data is represented through multiple distinct feature sets or "views" (e.g., text and image) [29]. Common approaches include co-training and multi-view representation learning [30]. The central idea is to leverage the complementary information in these views to improve performance, often by enforcing consistency or agreement across them. In contrast, our approach aims to exploit variability among representation in order to achieve a more representative set of examples, rather than achieving a single coherent view of the data.

## 3 OUR METHOD: *MERS*

The proposed method (see Fig. 1), called *Multi-Embedding Replay Selection (*MERS*)*, is intended to enhance any replay-based approach within the CIL (Class Incremental Learning) framework. It involves 2 main steps: (i) replace the buffer selection method with a graph-based method; (ii) expand the graph-based method to integrate a supervised and self-supervised embeddings. The method's primary advantage is expected to emerge in low memory buffer scenarios.

The optimization problem, which lies at the heart of the new method, can be shown to be a known variant of the k-coverage problem, which is defined as follows:

**Definition 1** (Element-Weighted Maximum $k$-Coverage with two groups). *Let $U$ be a universe of elements, partitioned into two disjoint subsets $U^1$ and $U^2$ such that $U = U^1 \cup U^2$ and $U^1 \cap U^2 = \emptyset$. Each element $e \in U^i$ is associated with a nonnegative weight $\alpha_i(e) \in \mathbb{R}_{\geq 0}$, where the weight functions $\alpha_1, \alpha_2$ may differ between the two groups.*

*Let $\mathcal{S} = \{S_1, S_2, \ldots, S_l\}$ be a family of subsets of $U$, and let $k \in \mathbb{N}$ be a budget parameter. For a subcollection $\mathcal{A} \subseteq \mathcal{S}$, define the* coverage weight *as*

$$\text{Coverage}(\mathcal{A}) = \sum_{e \in \bigcup_{S \in \mathcal{A}} S \cap U^1} \alpha_1(e) + \sum_{e \in \bigcup_{S \in \mathcal{A}} S \cap U^2} \alpha_2(e). \tag{1}$$

*The goal is to select a subcollection $\mathcal{A} \subseteq \mathcal{S}$ of size at most $k$ that maximizes* $\text{Covearge}(\mathcal{A})$.

The problem can be extended to multiple groups, corresponding to multiple embeddings. A natural greedy algorithm, which iteratively selects the set with the largest marginal gain in coverage, achieves $(1 - 1/e)$-approximation for this problem [25].

### NOTATIONS

Let $X = \{x_i\}_{i=1}^n$ represent a set of N data points, where $x_i \in \mathcal{X}$. For this dataset define the graph $G = (V, E)$, with vertices $V = \{v_i\}_{i=1}^n$ where $v_i \leftrightarrow x_i$, and edges $e_{i,j} = D(x_i, x_j)$ for some distance metric $D : \mathcal{X} \times \mathcal{X} \to \mathbb{R}_{\geq 0}$.

With multiple embeddings, each dataset can now be represented by a collection of graphs $\{V, E^{(m)}\}$, where $m \in [M]$ indexes the embeddings, $v_i \leftrightarrow x_i$ and $e_{i,j}^{(m)} = D(z_i^{(m)}, z_j^{(m)})$ for an embedding $f^{(m)} : \mathcal{X} \to \mathcal{Z}^{(m)}$.

**Definition 2** (Cover ball). *Fix $\delta > 0$, and consider an embedding $f^{(m)} : \mathcal{X} \to \mathcal{Z}^{(m)}$ where $z_x^{(m)} = f^{(m)}(x)$. Define*

$$B_\delta^{(m)}(x) = \{x' \in \mathcal{X} \mid D(z_{x'}^{(m)}, z_x^{(m)}) \leq \delta\}$$

$B_\delta^{(m)}(x)$ *denotes the set of points whose embedding lies inside the ball of radius $\delta$ centered at $x$ in embedding $m$[1].*

### 3.1 COVERAGE-BASED SELECTION

Our method is designed to enhance any coverage-based selection method within the framework of *replay-based CIL*. These methods aim to select a small, representative subset of the original dataset; their core rationale is to reduce the selection problem to one of maximizing graph coverage. Thus, the selection of subset of $b$ elements is reduced to max probability coverage as follows:

---

[1]Superscript $(m)$ can be omitted with a single embedding.

**Definition 3** (Max Probability Cover). *Fix $\delta > 0$, and obtain a subset $\mathcal{M} \subset X$ with $|\mathcal{M}| = b$ that maximizes the probability of the covered area:*

$$\mathcal{M} = \underset{L \subseteq X,\ |L|=b}{\arg\max}\ P\left(\bigcup_{x \in L} B_\delta(x)\right) \tag{2}$$

In *ProbCover* [27], the probability of the covered area is estimated by way of the empirical likelihood - $P(\bigcup_{x \in L} B_\delta(x)) = \left|\bigcup_{x \in L} B_\delta(x)\right|$. In other words, we seek a subset $\mathcal{M}$ for which the number of points in the original dataset that lie within distance $\delta$ of the points in $\mathcal{M}$ is as large as possible. Finally, in *MaxHerding* [3], the probability of the covered area is estimated with an RBF kernel, centered at each of the selected points in set $\mathcal{L}$.

### 3.2 Buffer selection, multiple embeddings

We begin by generalizing (2) to multiple embeddings:

**Definition 4** (Buffer selection, weighted coverage). *Obtain a subset of elements $\mathcal{M} \subset \mathbb{X}$, $|\mathcal{M}| = b$, that maximizes*

$$\mathcal{M} = \underset{L \subseteq X,\ |L|=b}{\arg\max} \sum_{m=1}^{M} \alpha_m \left|\bigcup_{x \in L} B_\delta^{(m)}(x)\right| \tag{3}$$

In (3), we seek a subset $\mathcal{M}$ that maximizes the coverage probability in all the embeddings. This is achieved by defining a notion of *weighted coverage*, which sums over the coverage in each embedding with weight $\alpha_m$. The weights reflect the relative importance (or effectiveness) of each embedding, and need to be determined by the algorithm.

The optimization problem in (3) is an instance of the *weighted maximum $k$-coverage with $m$ groups* (Def. 1), under the following correspondence: (i) $U^m$ corresponds to the set $X$ in embedding $m$, i.e., for every $x_i \in X$ there exists a corresponding $u_i^m \in U^m$. (ii) For each datapoint $x_i$ we associate the subset $S_i = \bigcup_m B_{\delta_m}^{(m)}(u_i^m)$, $\mathcal{S} = \{S_1, S_2, \ldots, S_m\}$.

### 3.3 Embedding alignment

In each coverage-based selection method described above, there is at least one parameter that captures the range of similarities in the data, and how well the data is partitioned. Each algorithm is sensitive in different ways to this parameter, and therefore its automatic evaluation from the data is crucial to the effectiveness of the method. This problem is exacerbated in our work, as we integrate multiple embedding spaces $\{\mathcal{E}^{(m)}\}_{m=1}^{M}$ that originate from distinct backbones and therefore may exhibit markedly different geometric characteristics. Therefore, a fixed one-size-fits-all solution is not likely to be effective.

For the purpose of embedding alignment, we use $k$-Nearest Neighbor ($k$-NN) density estimation, which is a non-parametric technique widely used in machine learning to estimate an unknown sample distribution without making any parametric assumptions. This model is used in [27; 3] for the generalization analysis of both *ProbCover* and *MaxHerding* because it depends exclusively on distances from a set of training examples and does not involve any additional inductive bias.

Specifically, when integrating either *ProbCover* or *MaxHerding* (defined in Section 3.1) into *MERS*, the relevant parameters are $\delta$ from Def. 2, and the bandwidth of the RBF kernel $\sigma$ respectively. In order to *adapt* each hyper-parameter to the statistics *within* the relevant embedding, we propose to use the median of $k$-NN distances for $\delta$ where $k$ is determined by the memory-aware ratio, and the median heuristic for $\sigma$. This allows the method to align the covering sets and kernel similarities with the true geometry and sparsity of $\mathcal{E}^{(m)}$.

This alignment is critical: it guarantees that *MaxHerding* selects representatives at the right granularity, and that *ProbCover* strikes an optimal balance between coverage and diversity, *regardless of which embedding space is active at any point in the stream*, as demonstrated in the ablation study (Section 6).

### 3.3.1  SELECTION OF $\delta$ IN `ProbCover`

We estimate the radius $\delta$ of each *Cover ball* (see Def. 2) by computing the median of the $k$-nearest neighbor ($k$-NN) distances for every feature vector in the current stream:

More specifically, let $\mathcal{M}_c = \{x \in X \mid y(x) = c\}$. For each $\mathbf{x}_i \in \mathcal{M}_c$, let $\mathcal{N}_K(\mathbf{x}_i)$ be the set of its $k$ nearest neighbors in $\mathcal{M}_c \setminus \mathbf{x}_i$. Compute the median distance from $\mathbf{x}_i$ to its $k$ nearest neighbors: $r_i := \mathrm{median}_{\mathbf{x}_j \in \mathcal{N}_K(\mathbf{x}_i)} \left\| \mathbf{x}_i - \mathbf{x}_j \right\|$. Finally, take the median over set $\{r_i\}$ and fix $\delta = \mathrm{median}_{i \in [N]} r_i$ to be the radius of $\mathcal{B}_\delta$ in *ProbCover*.

The neighborhood size $k$ is also a data-driven hyper-parameter that is being adapted to both the stream statistics and the class-specific memory budget: $K = \frac{|\mathcal{D}_c|}{\mathcal{M}_c}$, where $|\mathcal{D}_c|$ denotes the number of samples of class $c$ observed in the current episode, and $\mathcal{M}_c$ is the buffer capacity allocated to that class. This ratio partitions the feature space into as many covering regions as the buffer can hold: a larger buffer yields a finer subdivision (larger $k$), whereas a smaller buffer enforces coarser regions. For the Model based embedding, we fix $K = 1$ to enforce localized neighborhood selection. Because this representation space is denser, a smaller $\delta$ is required to prevent neighborhoods from overlapping excessively, thereby maintaining adequate sample diversity in the buffer.

### 3.3.2  BANDWIDTH SELECTION FOR THE RBF KERNEL IN `MaxHerding`

When *MaxHerding* is used for coverage-based selection, it employs the radial basis function (RBF) kernel $\kappa(\mathbf{x}, \mathbf{x}') = \exp\left(-\|\mathbf{x} - \mathbf{x}'\|^2 / (2\sigma^2)\right)$. Following the widely adopted *median heuristic* [13], we set the bandwidth $\sigma$ to the median cosine distances among all exemplars in the current episode.

### 3.3.3  WEIGHTING EACH EMBEDDING

Finally, we discuss the estimation of the vector of weights $\{\alpha_m\}$ defined in (3).

First, we recall the definition of the $k$-NN density estimation. Once again, let $\mathcal{M}_c = \{x \in X \mid y(x) = c\}$. For any $x \in \mathcal{M}_c$, let $\mathcal{N}_K^{(m)}(x)$ denote the set of its $k$ nearest neighbors in $\mathcal{M}_c \setminus \mathbf{x}$ in embedding $\mathcal{E}^{(m)}$. Let $\rho_k^{(m)}(x)$ denote the mean distance from $x$ to set $\mathcal{N}_K(\mathbf{x})$. In embedding $m$, the kNN density estimate at $x$ is defined as follows:

$$\widehat{f}_k^{(m)}(x) = \frac{k}{\rho_k^{(m)}(x)} \tag{4}$$

For embedding $m$, we now defined its weight as follows:

$$\alpha_m = \frac{\mathrm{median}(\widehat{f}_k^{(m)}(x))}{\mathrm{median}(\widehat{f}_1^{(m)}(x))} \tag{5}$$

The reasoning behind this definition is as follows: if two point clouds differ only by a scale factor, the distribution of $\alpha$ remains unchanged, resulting in $\alpha_1 = \alpha_2$. In practice, however, the supervised embedding $\mathcal{E}_{\mathrm{Model\ Based}}$ tends to exhibit *micro-clusters* - tightly grouped, nearly identical samples within a class - more so than the self-supervised embedding $\mathcal{E}_{\mathrm{self\text{-}supervised}}$. These geometric irregularities disrupt scale invariance: $\rho_1/\rho_k$ in $\mathcal{E}_{\mathrm{sup}}$ and deflate it in $\mathcal{E}_{\mathrm{self}}$, so that

$$\beta = \frac{\alpha_{\mathrm{Model\ Based}}}{\alpha_{\mathrm{self\text{-}supervised}}} > 1. \tag{6}$$

Our greedy algorithm maximizes the *weighted coverage score* defined in (3). Because the algorithm also enforces *diversity* through disjoint $k$-NN balls, dense supervised balls contain far fewer candidate edges than large self-supervised balls. Multiplying the supervised edge count by $\beta$ therefore equalizes the **effective area** (i.e. edge mass) that each selected point can cover, ensuring that the sampler does not over-represent the sparse self-supervised space and achieves a balanced, diverse subset across both embeddings.

## 3.4 PSEUDO-CODE

We propose two variants of our method *MERS*, that are distinguished by the coverage-based method they incorporate - *ProbCover* or *MaxHerding*. Pseudo-code for these 2 variants is provided in Algorithm 1 and Algorithm 2 respectively.

---

**Algorithm 1** *MERS ProbCover*

---

**Input**: Set $(X_m)$ of exemplars from $m$ embeddings, Memory buffer $\mathcal{M}$, Ball-size $\delta$
**Output**: $\mathcal{M}$

1: $B_\delta^{(m)}(x) = \left\{ x' \in \mathcal{X} \mid D\big(z_{x'}^{(m)}, z_x^{(m)}\big) \leq \delta \right\}$
2: $G = \left\{ V, E^{(m)} = \left\{ (x, x') : x' \in B_\delta^{(m)} \right\} \right\}$
3: **for** $i \in \{1, \ldots, b\}$ **do**
4:     $\mathcal{M}$ $\leftarrow$
    $argmax\left( \sum\limits_{m=1}^{M} \alpha_m \Big| \bigcup\limits_{x \in X_m} B_\delta^{(m)}(x) \Big| \right)$
5:     Remove all the incoming edges to covered vertices,
    $E \leftarrow E \setminus \left\{ (x, x') : x' \in B_\delta^{(m)} \right\}$
6: **end for**
7: **return** $\mathcal{M}$

---

**Algorithm 2** *MERS MaxHerding*

---

**Input**: Set $(X_m)$ of exemplars from $m$ embeddings, Memory buffer $\mathcal{M}$
**Output**: $\mathcal{M}$

1: Compute integrated kernel:
    $k(x, x') = \sum_{m=1}^{M} \alpha_m k_m\big(x^{(m)}, x'^{(m)}\big)$
2: $\mathcal{B} \leftarrow \varnothing$; $\mathbf{k} \in \mathbb{R}^{|\mathcal{C}|}$ with $k_i = 0$
3: **for** $b \in \{1, \ldots, B\}$ **do**
4:     Select
    $x_b^* = \arg \max\limits_{\tilde{x} \in \mathcal{C}} \frac{1}{|\mathcal{C}|} \sum\limits_{n=1}^{|\mathcal{C}|} \max\big(k(x_n, \tilde{x}) - k_n, 0\big)$
5:     Update $k_i \leftarrow \max\big(k(x_i, x_b^*), k_i\big)$
6:     $\mathcal{B} \leftarrow \mathcal{B} \cup \{x_b^*\}_{b=1}^{B}$
7:     $\mathcal{C} \leftarrow \mathcal{C} \setminus \{x_b^*\}_{b=1}^{B}$
8: **end for**

---

## 4 METHODOLOGY

In our empirical evaluation, we separately evaluate each of the two aforementioned variants of our method. We report 3 scenarios: (i) **SimCLR** *MERS* uses a single unsupervised embedding - the SimCLR method recalled below; (ii) **Model Based** *MERS* uses a single supervised embedding, obtained from the learned classifier; (iii) **integrated** *MERS* as defined in Algorithms 1-2, which integrate the two embeddings.

Our method is evaluated while enhancing 3 distinct experience replay continual learning algorithms, detailed in Section 4.1. It is compared to the vanilla version of each method, and to another method that can be used to enhance the buffer selection step, which is described in Section 4.2. The two coverage-based methods, employed by the variants of *MERS*, are expanded on in Section 4.3. Section 4.4 describes the two datasets used in our evaluation, following customary practice in the evaluation of CIL methods. Common evaluation metrics are described in Section 4.5.

**SimCLR [11]** is a self-supervised contrastive learning method that learns visual representations by leveraging data augmentation and a contrastive loss, without requiring any labeled data. The core idea is to train a neural network to recognize that different augmented views of the same image should have similar representations, while views of different images should be distinct. This is achieved by applying random augmentations to each image in the dataset, creating pairs of correlated views that serve as positive examples. All other images in the batch serve as implicit negatives, encouraging the model to distinguish between different inputs based solely on appearance.

**VICReg [6]** is a self-supervised learning method that combines three complementary regularization terms: (i) The invariance term encourages representations of different augmented views of the same image to be close, (ii) the variance term prevents representation collapse by ensuring each dimension has non-trivial variance across a batch, (iii) and the covariance term reduces redundancy by decorrelating feature dimensions. Together, these constraints produce stable and diverse embeddings that transfer well to downstream tasks, while simplifying training compared to contrastive approaches.

**DINO [9]** is a self-supervised distillation method where a student matches soft targets from a teacher on augmented views of the same image. The teacher is an exponential moving average (EMA) of the student, providing stable targets without labels. DINOv2 [19] extends this with Vision Transformers, a 142M-image dataset, and training refinements, producing versatile representations transferable to tasks like classification, retrieval, and clustering.

## 4.1 CONTINUAL LEARNING ALGORITHMS

We evaluated *MERS* with three rehearsal-based continual learning algorithms. First, **ER [21]** stores a subset of past examples in a memory buffer and replays them during training to reduce forgetting. Then, **ER-ACE [7]** extends ER by separating the loss contributions of new data and replayed samples. Finally, **MIR [1]** selects from the buffer the samples whose loss increases most after a gradient step on the current batch, focusing rehearsal on knowledge most at risk of forgetting.

## 4.2 ALTERNATIVE SELECTION STRATEGIES

Herding [26; 20] is one of the earliest exemplar selection strategies. It constructs a representative memory by sequentially selecting samples whose inclusion best approximates the class mean in feature space, ensuring that the chosen exemplars collectively act as a centroid for their class. Rainbow Memory [4] takes a complementary approach by explicitly balancing multiple selection criteria, such as diversity, uncertainty, and class balance, when building the memory buffer. TEAL [23] is a low-budget exemplar selection strategy for CIL. It clusters class samples in feature space and selects the most typical point from each cluster, ensuring both diversity and representativeness.

## 4.3 DIFFERENT COVERAGE STRATEGIES

***ProbCover* [27]** An active-learning algorithm that maximizes coverage under a small budget by building an $r$-neighborhood graph with radius $\delta$ and iteratively selecting the node with maximal uncovered degree. For continual learning, we adapt it by treating the memory buffer as the pool and the exemplar set as labeled data (see Section 3.4).

***MaxHerding* [3]** An active learning method that generalizes *ProbCover* by replacing hard $\delta$-ball coverage with a continuous kernel-based similarity measure. In its greedy variant, *MaxHerding* evaluates, at each step, a gain function that favors points in dense regions while penalizing redundancy with the current exemplar set, and then selects the sample with the highest gain.

## 4.4 DATASETS

Two datasets, that are commonly used to evaluate CIL methods, are used here: **(i) Split CIFAR-100**[10; 20], created by splitting CIFAR-100 and is divided into 10 episodes, each containing 10 different classes with 500 train images, and 100 test images. **(ii) Split TinyImageNet**, [16] created by splitting TinyImageNet and is divided into 10 episodes, each contain 20 different classes with 500 train images and 50 test images.

## 4.5 EVALUATION METRICS IN CIL

The **Average Accuracy** $(AA_t)$ is the mean accuracy over all tasks up to task $t$. The **Final Average Accuracy (FAA)** is the average accuracy after training on the last task $T$, i.e.,$FAA = AA_{t=T}$, which measures overall performance across all tasks. The **Any-time Average Accuracy (AAA)** is the mean of $AA_t$ across all $T$ tasks:$AAA = \frac{1}{T}\sum_{t=1}^{T} AA_t$.

# 5 EMPIRICAL RESULTS

## 5.1 MAIN RESULTS

In our empirical evaluation, we assess two variants of our *MERS* that rely on two different coverage based method, denoted *MERS ProbCover* and *MERS MaxHerding*. Each method is evaluated in 3 conditions: (i) constrained to use a single unsupervised SimCLR embedding; (ii) constrained to use only the supervised embedding derived from the classifier; (iii) allowed to exploit both embeddings as defined in Algorithms 1–2. These variants are used to enhance three existing experience replay continual learning algorithms, and are compared against both the vanilla versions of those algorithms and 3 alternative buffer selection methods - TEAL, Herding and Rainbow Memory - described in Section 4.2. To assess robustness to memory constraints, we varied the replay-buffer capacity across multiple sizes, from 100 to 1000 on the Split CIFAR-100 benchmark, and 200 to 6000 on the Split

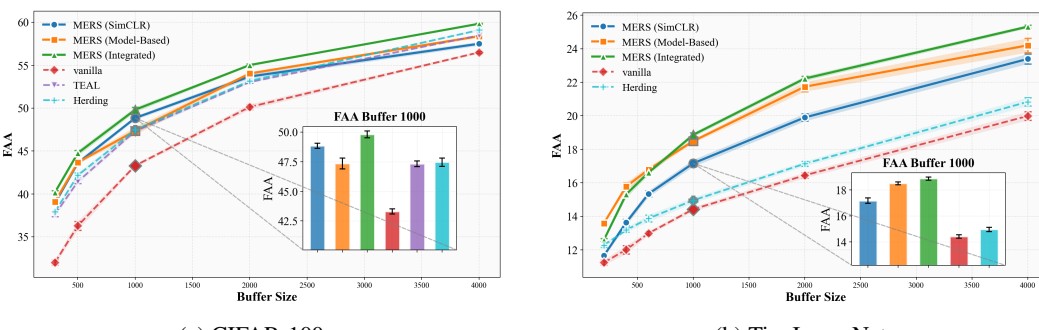

(a) CIFAR-100          (b) TinyImageNet

Figure 2: **MERS ProbCover**: FAA of ER-ACE as a function of $|\mathcal{M}|$, on CIFAR-100 (left) and TinyImageNet (right). Three variants of *MERS ProbCover* are evaluated and compared against alternative selection strategies.

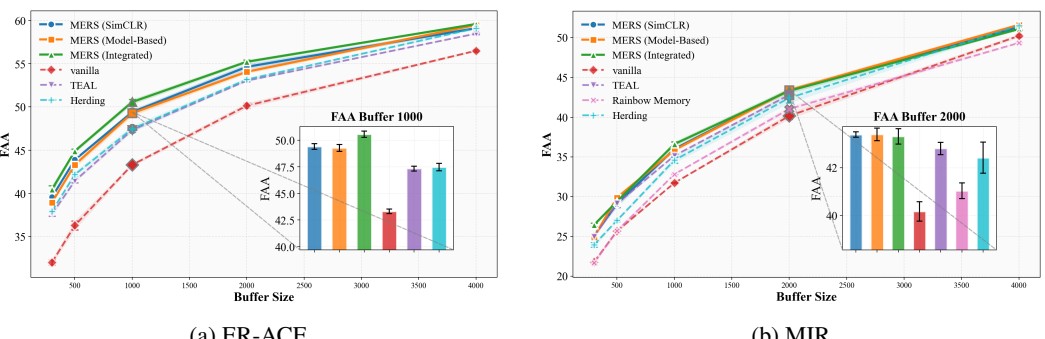

(a) ER-ACE          (b) MIR

Figure 3: **MERS MaxHerding**: FAA of ER-ACE (left) and MIR (right) on *CIFAR-100*, as a function of $|\mathcal{M}|$. Results with *MERS MaxHerding* are compared against other selection strategies.

TinyImageNet benchmark. Final Average Accuracy (FAA) is reported in Figs. 2–4, while the complete results, including FAA and AAA, are provided in Appendix A (see Tables 1–10).

## 5.2 IMPACT OF PRETRAINED VS. EPISODIC EMBEDDINGS ON *MERS*

Following the main results protocol (Section 5.1), we evaluate *MERS* with ER-ACE on Split CIFAR-100 across different buffer sizesusing embeddings beyond SimCLR. (i) **VICReg** is trained from scratch at each episode using only the current episode's training data, identical to the SimCLR protocol. (ii) **DINOv2** embeddings are extracted from a foundational model (Section 4). Results are presented in Fig. 5, with complete FAA and AAA tables reported in Appendix A.

## 5.3 DISCUSSION

Across every buffer size, experience replay base-method and dataset, *MERS ProbCover* achieves the most competitive results. Its integrated variant matches or exceeds its constrained variants,

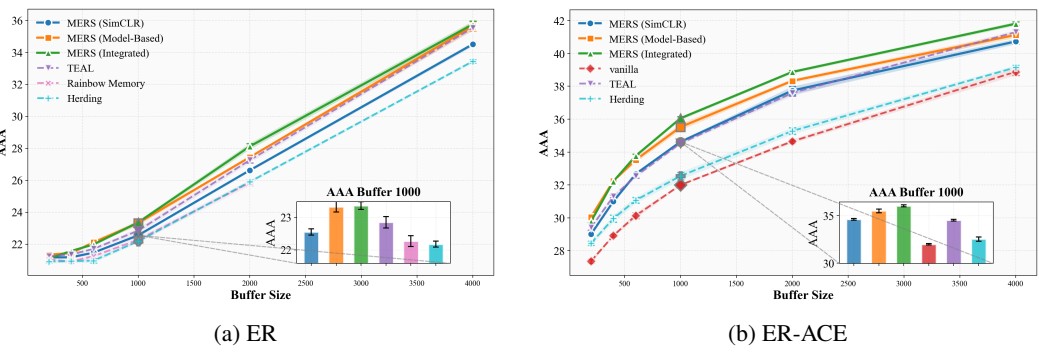

(a) ER          (b) ER-ACE

Figure 4: **MERS ProbCover**: AAA of ER (left) and ER-ACE (right) on *TinyImagenet*, Three variants of *MERS ProbCover* are evaluated, compared to other selection strategies

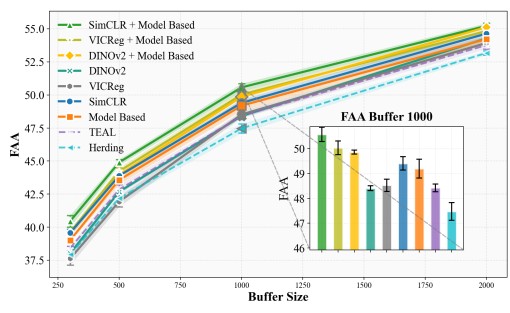 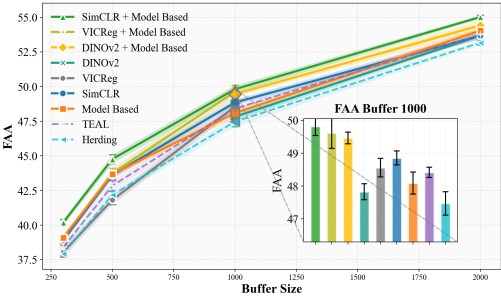

(a) *MERS MaxHerding*     (b) *MERS ProbCover*

Figure 5: FAA of *MERS MaxHerding* (left) and *MERS ProbCover* (right) with ER-ACE on *CIFAR-100* with different embeddings: SimCLR, VICReg and DINOv2

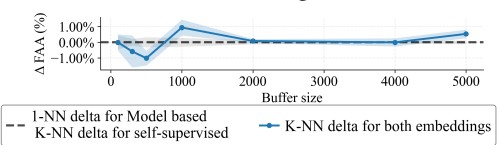 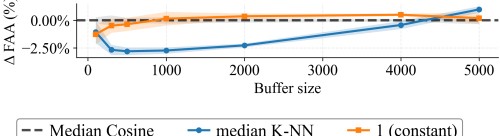

Figure 6: Improvements in FAA on CIFAR-100 as a function of $|\mathcal{M}|$ while varying *ProbCover*'s radius $\delta$.

Figure 7: Improvements in FAA on CIFAR-100 as a function of $|\mathcal{M}|$ while varying the RBF bandwidth $\sigma$ in *MaxHerding*.

an advantage that is most pronounced in the low-budget regime (up to 1000 exemplars), where it opens a clear gap over the 2 constrained variants. As $|\mathcal{M}|$ increases the gap indeed narrows, yet the integrated *MERS* still *retains first place*, sharing the highest score with one of the constrained variants. *MERS MaxHerding* shows a split pattern, staying ahead of the constrained variants across all buffer sizes only on CIFAR-100. Overall, *MERS* significantly outperforms either embedding in isolation, with the integrated design providing a distinct advantage under limited memory. In particular, episodic embeddings drive stronger task adaptation when memory is constrained.

# 6 ABLATION STUDY

We conducted targeted ablations to identify which design choices of our MERS are most critical:

**Adaptive *ProbCover* radius $\delta$:** Fig. 6 shows how varying $\delta$ affects *MERS ProbCover*. When used for active learning, the algorithm suffers from high sensitivity to $\delta$, as argued in [3]. We note that the optimal value of $\delta$ differs between Model based and self-supervised embeddings, reflecting the distinct statistical properties of supervised versus contrastive representations.

**RBF bandwidth $\sigma$ in *MaxHerding*** We tested three settings for $\sigma$: (i) median cosine distances, (ii) $\sigma = 1$, and (iii) median $k$-NN distances. On CIFAR-100, (i) and (iii) coincide, while the constant value reduces FAA by $\approx \mathbf{1\%}$ in the small-buffer regime. As (i) is dataset-agnostic and robust across budgets, we adopt it as the default.

We conducted an ablation study on the embedding weight $\alpha$ using different density estimators. The results show a slight improvement when using the $\alpha$ defined in (5), as reported in Appendix A.

# 7 SUMMARY

We present Multi-Embedding Replay Selection (*MERS*), a plug-and-play sampler for replay-based continual learning that merges supervised and self-supervised feature spaces in a complementary manner. By building $k$-NN coverage graphs in each space, re-scaling them with density-aware weights, and greedily selecting exemplars that maximize a combined coverage score, *MERS* fills both class-discriminative and invariant regions of the data manifold. Across Split CIFAR-100 and Split TinyImageNet, it boosts final-average accuracy over single-embedding baselines when memory is tight, all without increasing the buffer size or changing model parameters. The method is plug-and-play, incurs only double selection-time overhead and self-supervised training. The approach opens avenues for dynamic, task-aware embedding integration in future work.

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

## A APPENDIX

### A.1 USE OF LLMS

A large language model (ChatGPT, GPT-5 by OpenAI) was employed solely for minor editorial assistance. All methodological design, experimental results, and scientific conclusions are entirely the authors' own.

## A.2 TIME AND SPACE COMPLEXITY OF *MERS*

We analyse the computational cost under the standard setting in which the selection strategy is invoked *once per training episode*. Let $n$ be the number of examples from the current episode that belong to class $c$, $M$ the number of distinct embedding spaces, $d$ the dimensionality of each embedding, and $b$ the class-wise memory-buffer budget ( the number of items that $|\mathcal{M}|$ may store for class $c$).

**Self-supervised stage.** During every episode, *MERS* is called exactly once. Running SimCLR for $E_{\text{ssl}}$ epochs on $A = 2$ views of the $n$ episode images costs

$$T_{\text{SimCLR}} = O(E_{\text{ssl}}\, A\, n\, P)$$

with $P$ trainable parameters. Self-supervised training consumes

$$S_{\text{SimCLR}} = O(P + s\, f)$$

space model parameters $P$ plus the current batch's $s$ activations of size $f$, and the batch size $s$. The SimCLR weights are discarded after each episode, persistent memory is dominated by the replay images.

### A.2.1 *MERS ProbCover*

The algorithm consists of two stages:

**(i) Ball-graph construction.** For every embedding $m \in \{1,\ldots,M\}$ we compute all pairwise cosine distances in $\mathbb{R}^d$ to obtain the $\delta$-neighbourhoods $B_\delta^{(m)}(x)$. This step costs $T_{\text{graph}} = O(M\, n^2\, \max\{d,b\})$ and stores $S_{\text{graph}} = O(M\, n^2)$ adjacency edges.

**(ii) Greedy covering.** Across $b$ iterations we repeatedly pick the vertex that covers the largest number of still-uncovered neighbours. The work per iteration yields $T_{\text{cover}} = O(|E| + b\, n) \subseteq O(M\, n^2 + b\, n)$.

**Overall complexity.**

$$T_{\text{MERS-}\textit{ProbCover}} = O(Mn^2 \max d, b),$$

$$S_{\text{MERS-}\textit{ProbCover}} = O(Mn^2),$$

The original ProbCover analysis [27] reports a running time of $O(n^2 \max\{d,b\})$. Our derivation shows that the multi-embedding extension, *MERS –ProbCover*, retains the same quadratic dependence on $n$ and on $\max d, b$, differing only by the multiplicative factor $M$ (which equals 2 in all of our experiments).

**(ii) Greedy *MaxHerding* selection.**

**(i) Integrated-kernel construction.** We assemble the Gram matrix

$$K_{ij} = k(x_i, x_j) = \sum_{m=1}^{M} \alpha_m\, k_m\big(x_i^{(m)}, x_j^{(m)}\big).$$

Forming its $\frac{1}{2}n(n-1)$ entries costs

$$T_{\text{kernel}} = O(mn^2 d), \qquad S_{\text{kernel}} = O(n^2).$$

**(ii) Greedy selection.** Each of the $b$ iterations scans all candidates ($\leq n$) and exploits the precomputed kernel:

$$T_{\text{MaxHerding}} = O(b\, n^2), \qquad S_{\text{MaxHerding}} = O(n).$$

**Overall complexity.**

$$T_{\text{MERS–}\textit{MaxHerding}} = O\big(mn^2(d+b)\big),$$

$$S_{\text{MERS–}\textit{MaxHerding}} = O\big(n^2 + nd\big).$$

### A.3 HYPERPARAMETERS

#### A.3.1 CLASSIFICATION MODEL

we employ a ResNet-18 backbone trained for 100 epochs with a batch size of 10. The **ER-ACE** configuration begins with a learning rate of 0.01. The **ER** and **MIR** configuration begins with a learning rate of 0.1, for all configurations, SGD optimization includes Nesterov momentum of 0.9 and weight decay 0.0002. The learning rate is decayed by a factor of 0.3 every 66 epochs. All experiments were run with five random seeds (0-4).

#### A.3.2 CLASS ORDER

We follow the canonical class order for each benchmark: Split CIFAR-100 uses classes $[1 \dots 100]$, and Split TinyImageNet uses classes $[1 \dots 200]$.

#### A.3.3 SELF-SUPERVISED TRAINING

Our SimCLR implementation is adapted from solo-learn[12], and is available in the source code. The self-supervised model is trained on the images observed in the current episode only, never on the full dataset.

#### A.3.4 FEATURE NORMALIZATION

Each feature vector is divided by its $\ell_2$ norm, yielding unit-norm representations. Similarities are therefore computed with the cosine distance.

### A.4 COMPUTE RESOURCES

Each experiment trained deep-learning models on GPUs, consuming up to 22 GB of GPU memory and no more than 20 GB of system RAM.

### A.5 SOURCE CODE

The complete source code is provided in the supplementary ZIP file and will be publicly released on GitHub upon acceptance. The source code includes a README that lists the commands required to reproduce all of the experiments described in this paper.

### A.6 MAIN RESULTS TABLES

The tables 1- 10 presents the complete tables underlying Figs. 2– 4, evaluated with both the FAA and AAA metrics.

### A.7 TRANSFER LEARNING TABLES

The tables 11- 12 presents the complete tables underlying Fig. 5, evaluated with both the FAA and AAA metrics.

Table 1: **Final Averaged Accuracy (FAA)** On **CIFAR-100** averaged over 5 independent runs (mean $\pm$ standard error). For each buffer size, the best accuracy is in bold; results within the standard error of the best are also bolded.

(a) **CIFAR-100** with the **ER-ACE** framework.

| $|\mathcal{M}|$ | ER-ACE (vanilla) Model Based | MERS ProbCover | | | MERS MaxHerding | | |
|---|---|---|---|---|---|---|---|
| | | Model Based | SimCLR | MERS | Model Based | SimCLR | MERS |
| 100 | 21.80 $\pm 0.34$ | 29.88 $\pm 0.24$ | 28.61 $\pm 0.21$ | 29.61 $\pm 0.43$ | 29.79 $\pm 0.25$ | 30.00 $\pm 0.32$ | **30.99** $\pm 0.29$ |
| 300 | 32.01 $\pm 0.30$ | 39.09 $\pm 0.22$ | 38.92 $\pm 0.21$ | **40.19** $\pm 0.22$ | 38.93 $\pm 0.08$ | 39.56 $\pm 0.28$ | **40.45** $\pm 0.44$ |
| 500 | 36.29 $\pm 0.52$ | 43.68 $\pm 0.17$ | 43.61 $\pm 0.28$ | **44.77** $\pm 0.32$ | 43.31 $\pm 0.34$ | 43.89 $\pm 0.11$ | **44.90** $\pm 0.23$ |
| 1000 | 43.30 $\pm 0.21$ | 47.36 $\pm 0.46$ | 48.85 $\pm 0.21$ | 49.82 $\pm 0.28$ | 49.28 $\pm 0.33$ | 49.40 $\pm 0.27$ | **50.57** $\pm 0.29$ |
| 2000 | 50.14 $\pm 0.30$ | 54.05 $\pm 0.18$ | 53.69 $\pm 0.27$ | **55.03** $\pm 0.13$ | 54.07 $\pm 0.22$ | 54.64 $\pm 0.33$ | **55.22** $\pm 0.20$ |
| 4000 | 56.50 $\pm 0.13$ | 58.38 $\pm 0.17$ | 57.51 $\pm 0.22$ | **59.84** $\pm 0.06$ | 59.48 $\pm 0.17$ | 59.11 $\pm 0.16$ | 59.59 $\pm 0.12$ |
| 5000 | 58.28 $\pm 0.26$ | 60.08 $\pm 0.08$ | 58.61 $\pm 0.17$ | **61.07** $\pm 0.12$ | 60.33 $\pm 0.12$ | 60.28 $\pm 0.17$ | 60.66 $\pm 0.35$ |

(b) **CIFAR-100** with the **ER** framework.

| $|\mathcal{M}|$ | ER (vanilla) Model Based | MERS ProbCover | | | MERS MaxHerding | | |
|---|---|---|---|---|---|---|---|
| | | Model Based | SimCLR | MERS | Model Based | SimCLR | MERS |
| 100 | 10.07 $\pm 0.13$ | **11.71** $\pm 0.06$ | 10.81 $\pm 0.04$ | 11.30 $\pm 0.10$ | 11.28 $\pm 0.08$ | 10.91 $\pm 0.07$ | **11.76** $\pm 0.16$ |
| 300 | 13.25 $\pm 0.10$ | **18.29** $\pm 0.28$ | 16.83 $\pm 0.17$ | **18.45** $\pm 0.32$ | 17.56 $\pm 0.14$ | 17.65 $\pm 0.35$ | **18.36** $\pm 0.24$ |
| 500 | 17.69 $\pm 0.30$ | **23.61** $\pm 0.26$ | 21.80 $\pm 0.20$ | **23.64** $\pm 0.36$ | 23.23 $\pm 0.29$ | 22.59 $\pm 0.17$ | **23.45** $\pm 0.35$ |
| 1000 | 26.04 $\pm 0.24$ | 31.99 $\pm 0.28$ | 31.29 $\pm 0.36$ | 33.13 $\pm 0.35$ | 32.46 $\pm 0.26$ | 32.11 $\pm 0.19$ | **33.31** $\pm 0.08$ |
| 2000 | 38.30 $\pm 0.23$ | 42.55 $\pm 0.52$ | 42.91 $\pm 0.26$ | **43.68** $\pm 0.22$ | 42.37 $\pm 0.29$ | 41.83 $\pm 0.96$ | **43.85** $\pm 0.25$ |
| 4000 | 50.63 $\pm 0.10$ | 53.12 $\pm 0.08$ | 53.39 $\pm 0.19$ | **53.97** $\pm 0.19$ | 53.34 $\pm 0.22$ | 52.73 $\pm 0.20$ | 53.21 $\pm 0.29$ |
| 5000 | 53.86 $\pm 0.37$ | 56.07 $\pm 0.26$ | 55.87 $\pm 0.49$ | **56.27** $\pm 0.14$ | 55.91 $\pm 0.31$ | 55.66 $\pm 0.39$ | 56.01 $\pm 0.17$ |

(c) **CIFAR-100** with the **MIR** framework.

| $|\mathcal{M}|$ | MIR (vanilla) Model Based | MERS ProbCover | | | MERS MaxHerding | | |
|---|---|---|---|---|---|---|---|
| | | Model Based | SimCLR | MERS | Model Based | SimCLR | MERS |
| 100 | 17.80 $\pm 0.32$ | 19.80 $\pm 0.28$ | 19.48 $\pm 0.25$ | **20.32** $\pm 0.14$ | 19.89 $\pm 0.15$ | 19.71 $\pm 0.46$ | 19.78 $\pm 0.23$ |
| 300 | 21.78 $\pm 0.17$ | 26.03 $\pm 0.25$ | 24.63 $\pm 0.31$ | 25.70 $\pm 0.20$ | 24.93 $\pm 0.36$ | 25.13 $\pm 0.20$ | **26.46** $\pm 0.22$ |
| 500 | 25.68 $\pm 0.24$ | 29.40 $\pm 0.12$ | 28.88 $\pm 0.15$ | **29.66** $\pm 0.20$ | **29.88** $\pm 0.26$ | 29.10 $\pm 0.44$ | 29.41 $\pm 0.22$ |
| 1000 | 31.74 $\pm 0.06$ | 35.36 $\pm 0.42$ | 34.85 $\pm 0.28$ | 35.89 $\pm 0.31$ | 35.88 $\pm 0.24$ | 36.01 $\pm 0.23$ | **36.62** $\pm 0.29$ |
| 2000 | 40.17 $\pm 0.41$ | **43.27** $\pm 0.16$ | 42.85 $\pm 0.27$ | **43.50** $\pm 0.24$ | **43.40** $\pm 0.26$ | **43.40** $\pm 0.11$ | 43.32 $\pm 0.33$ |
| 4000 | 50.23 $\pm 0.20$ | 51.29 $\pm 0.29$ | 51.31 $\pm 0.31$ | **51.74** $\pm 0.35$ | **51.61** $\pm 0.21$ | 51.35 $\pm 0.25$ | 51.09 $\pm 0.16$ |
| 5000 | 52.56 $\pm 0.34$ | 53.53 $\pm 0.29$ | 53.62 $\pm 0.22$ | **54.17** $\pm 0.40$ | **53.88** $\pm 0.27$ | 53.74 $\pm 0.28$ | 53.70 $\pm 0.39$ |

## A.8 ABLATION STUDY

Fig. 8 presents an ablation study on the effect of the embedding weight parameter $\alpha$ when using the median K-NN density defined in Eq. 4, applied to *MERS ProbCover* on CIFAR-100 under the ER-ACE setting. The results indicate a slight but consistent improvement when using the formulation of $\alpha$ given in Eq. 5.

We further investigate the influence of the hyperparameter $\delta$ in *MERS ProbCover*. Fig. 9 reports results when using a single embedding (either SimCLR or model-based). In both cases, the values of $\delta$ defined in Subsection 3.3.1 yield the best performance, confirming their suitability.

Table 2: **Final Averaged Accuracy (FAA)**, on **TinyImageNet** averaged over 5 independent runs (mean ± standard error). For each buffer size, the best accuracy is in bold; results within the standard error of the best are also bolded.

(a) **TinyImageNet** with the **ER-ACE** framework.

| $|\mathcal{M}|$ | ER (vanilla) Model Based | MERS ProbCover Model Based | SimCLR | MERS | MERS MaxHerding Model Based | SimCLR | MERS |
|---|---|---|---|---|---|---|---|
| 200 | 11.24 ±0.16 | **13.59** ±0.04 | 11.66 ±0.12 | 12.59 ±0.10 | 12.86 ±0.09 | 12.57 ±0.20 | 12.67 ±0.31 |
| 400 | 12.01 ±0.25 | **15.79** ±0.23 | 13.63 ±0.19 | 15.32 ±0.21 | 14.62 ±0.22 | 14.76 ±0.22 | 15.02 ±0.17 |
| 600 | 12.99 ±0.12 | **16.79** ±0.17 | 15.34 ±0.18 | 16.61 ±0.14 | 15.75 ±0.20 | 15.88 ±0.10 | 16.29 ±0.13 |
| 1000 | 14.42 ±0.13 | 18.49 ±0.11 | 17.16 ±0.21 | **18.86** ±0.13 | 18.06 ±0.20 | 17.33 ±0.17 | 17.73 ±0.21 |
| 2000 | 16.45 ±0.17 | 21.73 ±0.29 | 19.90 ±0.22 | **22.22** ±0.14 | 20.70 ±0.23 | 19.96 ±0.18 | 20.36 ±0.19 |
| 4000 | 19.99 ±0.26 | 24.20 ±0.42 | 23.39 ±0.30 | **25.32** ±0.10 | 24.27 ±0.24 | 22.99 ±0.11 | 23.78 ±0.21 |
| 6000 | 23.14 ±0.28 | 26.36 ±0.38 | 26.75 ±0.15 | **27.88** ±0.13 | 26.83 ±0.24 | 25.65 ±0.30 | 26.54 ±0.23 |

(b) **TinyImageNet** with the **ER** framework.

| $|\mathcal{M}|$ | ER (vanilla) Model Based | MERS ProbCover Model Based | SimCLR | MERS | MERS MaxHerding Model Based | SimCLR | MERS |
|---|---|---|---|---|---|---|---|
| 200 | 6.65 ±0.04 | 6.69 ±0.05 | 6.72 ±0.10 | 6.71 ±0.08 | **6.84** ±0.06 | 6.68 ±0.04 | 6.70 ±0.08 |
| 400 | 6.34 ±0.07 | **6.56** ±0.09 | 6.44 ±0.04 | **6.56** ±0.03 | 6.52 ±0.03 | 6.45 ±0.04 | 6.51 ±0.02 |
| 600 | 6.18 ±0.09 | **6.53** ±0.03 | 6.37 ±0.06 | **6.58** ±0.06 | **6.54** ±0.04 | 6.43 ±0.07 | 6.45 ±0.05 |
| 1000 | 6.23 ±0.03 | **7.05** ±0.06 | 6.53 ±0.10 | 6.80 ±0.07 | 6.81 ±0.05 | 6.75 ±0.06 | 6.63 ±0.08 |
| 2000 | 7.41 ±0.12 | 9.11 ±0.10 | 8.51 ±0.13 | **9.26** ±0.13 | 8.77 ±0.16 | 8.38 ±0.12 | 8.56 ±0.13 |
| 4000 | 11.87 ±0.15 | 15.68 ±0.29 | 14.53 ±0.26 | **15.94** ±0.20 | 15.05 ±0.35 | 13.75 ±0.15 | 14.72 ±0.30 |
| 6000 | 18.70 ±0.39 | 21.71 ±0.33 | 20.36 ±0.32 | **22.24** ±0.35 | 20.87 ±0.42 | 20.08 ±0.13 | 20.47 ±0.37 |

| $|\mathcal{M}|$ | Herding Model Based | TEAL Model Based | Rainbow Memory Model Based |
|---|---|---|---|
| 100 | 19.38 ±0.04 | **19.97** ±0.30 | 17.54 ±0.32 |
| 300 | 23.95 ±0.30 | **24.97** ±0.24 | 21.68 ±0.29 |
| 500 | 27.02 ±0.16 | **29.15** ±0.23 | 25.69 ±0.33 |
| 1000 | 34.60 ±0.37 | **35.16** ±0.15 | 32.83 ±0.06 |
| 2000 | 42.42 ±0.66 | **42.81** ±0.25 | 41.03 ±0.33 |

Table 3: FAA on **CIFAR-100** with the **MIR** framework. Comparison of Herding, TEAL, and Rainbow Memory across different buffer sizes

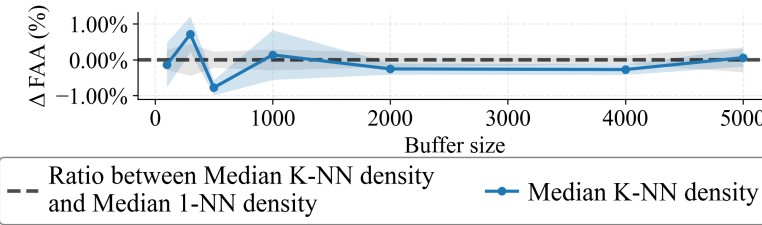

Figure 8: *MERS MaxHerding*. Ablation of the embedding weight $\alpha$ using K-NN density estimators on CIFAR-100 with ER-ACE. The baseline corresponds to Eq. (5), and a minor but consistent improvement is observed with this weighting.

Table 4: FAA on **CIFAR-100** with the **ER-ACE** framework. Comparison of Herding and TEAL Memory across different buffer sizes

| | Herding | TEAL |
|---|---|---|
| $|\mathcal{M}|$ | Model Based | Model Based |
| 100 | 29.93 $\pm 0.31$ | 29.67 $\pm 0.13$ |
| 300 | 37.92 $\pm 0.13$ | 38.40 $\pm 0.16$ |
| 500 | 42.18 $\pm 0.27$ | 42.84 $\pm 0.38$ |
| 1000 | 47.47 $\pm 0.35$ | 48.42 $\pm 0.16$ |
| 2000 | 53.17 $\pm 0.17$ | 53.76 $\pm 0.29$ |

Table 5: FAA on **CIFAR-100** with the **ER** framework. Comparison of Herding, TEAL, and Rainbow Memory across different buffer sizes

| | Herding | TEAL | Rainbow Memory |
|---|---|---|---|
| $|\mathcal{M}|$ | Model Based | Model Based | Model Based |
| 100 | 10.86 | 11.84 $\pm 0.12$ | 10.15 $\pm 0.10$ |
| 300 | 16.13 | 17.06 $\pm 0.13$ | 13.46 $\pm 0.10$ |
| 500 | 20.21 | 22.49 $\pm 0.20$ | 16.98 $\pm 0.60$ |
| 1000 | 30.01 | 31.92 $\pm 0.43$ | 26.72 $\pm 0.17$ |
| 2000 | 41.62 | 42.22 $\pm 0.51$ | 38.40 $\pm 0.22$ |

Table 6: FAA on **TinyImageNet** with the **ER-ACE** framework. Comparison of Herding, TEAL, and Rainbow Memory across different buffer sizes

| | Herding | TEAL |
|---|---|---|
| $|\mathcal{M}|$ | Model Based | Model Based |
| 200 | 12.27 $\pm 0.12$ | **13.32** $\pm 0.29$ |
| 400 | 13.18 $\pm 0.12$ | **14.91** $\pm 0.15$ |
| 600 | 13.88 $\pm 0.21$ | **15.66** $\pm 0.01$ |
| 1000 | 14.95 $\pm 0.17$ | **17.35** $\pm 0.17$ |
| 2000 | 17.14 $\pm 0.14$ | **20.11** $\pm 0.38$ |

Table 7: FAA on **TinyImageNet** with the **ER** framework. Comparison of Herding, TEAL, and Rainbow Memory across different buffer sizes

| | Herding | TEAL | RM |
|---|---|---|---|
| $|\mathcal{M}|$ | Model Based | Model Based | Model Based |
| 200 | 6.60 $\pm 0.08$ | 6.68 $\pm 0.03$ | **6.78** $\pm 0.03$ |
| 400 | 6.33 $\pm 0.03$ | 6.49 $\pm 0.03$ | 6.31 $\pm 0.11$ |
| 600 | 6.09 $\pm 0.05$ | 6.50 $\pm 0.06$ | 6.38 $\pm 0.03$ |
| 1000 | 6.18 $\pm 0.08$ | 6.68 $\pm 0.07$ | 6.43 $\pm 0.10$ |
| 2000 | 7.51 $\pm 0.11$ | 8.66 $\pm 0.10$ | 7.79 $\pm 0.41$ |

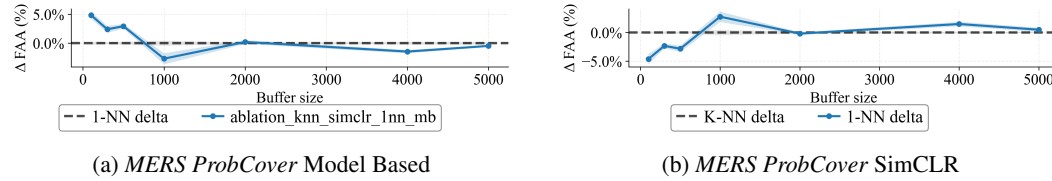

(a) *MERS ProbCover* Model Based          (b) *MERS ProbCover* SimCLR

Figure 9: *MERS ProbCover*. Ablation study of the hyperparameter $\delta$ on CIFAR-100 with ER-ACE, using a single embedding. Results are shown separately for (a) model-based embeddings and (b) SimCLR embeddings.

Table 8: **Average Accumulated Accuracy (AAA)**, averaged over 5 independent runs (mean ± standard error). For each buffer size, the best aaa is in bold; results within the standard error of the best are also bolded.

(a) **TinyImageNet** with the **ER-ACE** framework (AAA).

| | ER (vanilla) | *MERS ProbCover* | | | *MERS MaxHerding* | | |
|---|---|---|---|---|---|---|---|
| $\|\mathcal{M}\|$ | Model Based | Model Based | SimCLR | *MERS* | Model Based | SimCLR | *MERS* |
| 200 | 27.37 ±0.12 | **30.05** ±0.18 | 29.01 ±0.14 | 29.78 ±0.08 | 29.51 ±0.13 | 29.48 ±0.16 | 29.64 ±0.18 |
| 400 | 28.92 ±0.08 | **32.25** ±0.08 | 31.00 ±0.11 | **32.22** ±0.09 | 31.58 ±0.13 | 31.91 ±0.19 | 32.03 ±0.05 |
| 600 | 30.14 ±0.17 | 33.52 ±0.15 | 32.67 ±0.11 | **33.77** ±0.11 | 32.87 ±0.12 | 32.92 ±0.14 | 33.43 ±0.15 |
| 1000 | 32.00 ±0.07 | 35.52 ±0.19 | 34.62 ±0.08 | **36.05** ±0.10 | 35.25 ±0.16 | 34.76 ±0.13 | 34.81 ±0.06 |
| 2000 | 34.65 ±0.12 | 38.33 ±0.11 | 37.74 ±0.25 | **38.87** ±0.09 | 38.16 ±0.12 | 37.12 ±0.11 | 37.74 ±0.17 |
| 4000 | 38.88 ±0.21 | 41.12 ±0.17 | 40.72 ±0.16 | **41.80** ±0.11 | 40.98 ±0.19 | 40.37 ±0.09 | 40.91 ±0.12 |
| 6000 | 41.24 ±0.19 | 43.08 ±0.13 | 43.26 ±0.21 | **43.75** ±0.10 | 42.94 ±0.10 | 42.09 ±0.32 | 42.49 ±0.13 |

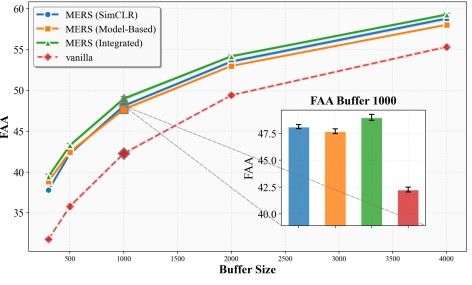 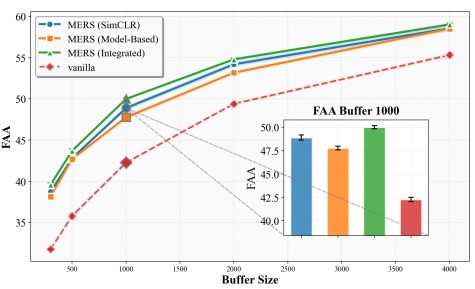

(a) *MERS ProbCover*  (b) *MERS MaxHerding*

Figure 11: FAA of ER-ACE on with *MERS ProbCover* (left) and *MERS MaxHerding* (right) is shown, as a function of the replay-buffer size ($|M|$).The class order is generated with **seed 42**, as described in A.9. Three variants of *MERS ProbCover* are evaluated (see text), and are being compared to the original ER-ACE (vanilla).

## A.9 ROBUSTNESS TO EPISODE CLASS ORDER IN CONTINUAL LEARNING

As in the experiments presented in Tables 1–2, we repeated them using different episode Class orders, defined by seeds 42 and 35. Below are the Final Averaged Accuracy and the Anytime Averaged Accuracy for seed 42: Tables 13, 15 and for seed 35: Tables 14, 16. and the results are analyzed in Fig. 10- 11

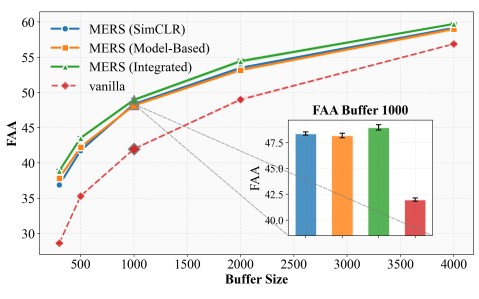 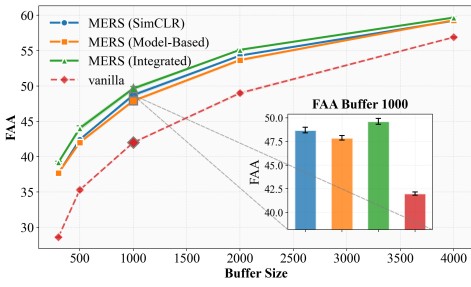

(a) *MERS ProbCover*  (b) *MERS MaxHerding*

Figure 10: FAA of ER-ACE on with *MERS ProbCover* (left) and *MERS MaxHerding* (right) is shown, as a function of the replay-buffer size ($|M|$).The class order is generated with **seed 35**, as described in A.9. Three variants of *MERS ProbCover* are evaluated (see text), and are being compared to the original ER-ACE (vanilla).

Table 9: **Average Accumulated Accuracy (AAA)**, averaged over 5 independent runs (mean ± standard error). For each buffer size, the best aaa is in bold; results within the standard error of the best are also bolded.

(a) **CIFAR-100** with the **ER-ACE** framework (AAA).

| $|\mathcal{M}|$ | ER (vanilla) Model Based | MERS ProbCover | | | MERS MaxHerding | | |
|---|---|---|---|---|---|---|---|
| | | Model Based | SimCLR | MERS | Model Based | SimCLR | MERS |
| 100 | 41.31 ±0.30 | 46.91 ±0.14 | 46.83 ±0.17 | 47.30 ±0.27 | 47.01 ±0.15 | 47.80 ±0.12 | **48.13** ±0.21 |
| 300 | 49.90 ±0.28 | 54.44 ±0.17 | 54.80 ±0.29 | 55.17 ±0.17 | 54.59 ±0.13 | 55.19 ±0.08 | **56.10** ±0.17 |
| 500 | 53.72 ±0.20 | 58.15 ±0.22 | 58.23 ±0.18 | 58.64 ±0.31 | 57.99 ±0.26 | 58.82 ±0.35 | **59.02** ±0.20 |
| 1000 | 58.88 ±0.09 | 61.87 ±0.24 | 62.18 ±0.34 | 63.00 ±0.28 | 62.57 ±0.17 | 62.85 ±0.35 | **63.44** ±0.26 |
| 2000 | 64.21 ±0.35 | 66.44 ±0.21 | 65.97 ±0.21 | 67.06 ±0.17 | 66.41 ±0.15 | 66.87 ±0.15 | **67.59** ±0.21 |
| 4000 | 68.98 ±0.11 | 70.27 ±0.07 | 68.84 ±0.24 | **70.64** ±0.13 | 70.43 ±0.21 | 69.46 ±0.32 | 70.07 ±0.09 |
| 5000 | 70.77 ±0.33 | 71.47 ±0.10 | 68.86 ±0.20 | **71.93** ±0.17 | 71.00 ±0.18 | 71.18 ±0.34 | 71.59 ±0.29 |

(b) **CIFAR-100** with the **ER** framework (AAA).

| $|\mathcal{M}|$ | ER (vanilla) Model Based | MERS ProbCover | | | MERS MaxHerding | | |
|---|---|---|---|---|---|---|---|
| | | Model Based | SimCLR | MERS | Model Based | SimCLR | MERS |
| 100 | 28.68 ±0.15 | 30.93 ±0.23 | 30.10 ±0.10 | 31.05 ±0.13 | 30.97 ±0.11 | 30.77 ±0.10 | **31.49** ±0.06 |
| 300 | 34.58 ±0.24 | 39.22 ±0.16 | 38.58 ±0.24 | 39.81 ±0.17 | 38.86 ±0.14 | 38.72 ±0.43 | **40.59** ±0.17 |
| 500 | 40.41 ±0.30 | 44.97 ±0.18 | 44.08 ±0.33 | 45.30 ±0.44 | 44.71 ±0.29 | 45.12 ±0.28 | **45.89** ±0.18 |
| 1000 | 49.84 ±0.45 | 52.99 ±0.37 | 53.07 ±0.28 | 53.91 ±0.24 | 53.90 ±0.10 | 53.71 ±0.40 | **54.40** ±0.13 |
| 2000 | 59.89 ±0.15 | 61.63 ±0.48 | **62.59** ±0.15 | 62.03 ±0.27 | 61.49 ±0.30 | 61.60 ±0.49 | **62.59** ±0.25 |
| 4000 | 68.41 ±0.09 | 69.13 ±0.09 | 69.53 ±0.13 | **70.18** ±0.16 | 69.13 ±0.24 | 68.83 ±0.18 | 69.05 ±0.32 |
| 5000 | 70.04 ±0.13 | 71.19 ±0.19 | 70.71 ±0.32 | **71.45** ±0.06 | 70.64 ±0.29 | 71.00 ±0.21 | 70.77 ±0.16 |

(c) **CIFAR-100** with the **ER-ACE** framework (AAA).

| $|\mathcal{M}|$ | Herding Model Based | TEAL Model Based | RM Model Based |
|---|---|---|---|
| 100 | 46.88 ±0.17 | 42.91 ±0.13 | 10.73 ±0.03 |
| 300 | 53.75 ±0.21 | 53.13 ±0.19 | 10.77 ±0.04 |
| 500 | 57.09 ±0.23 | 56.80 ±0.14 | 10.69 ±0.06 |
| 1000 | 61.67 ±0.29 | 61.10 ±0.28 | 10.75 ±0.01 |
| 2000 | 66.47 ±0.13 | 65.66 ±0.24 | 10.79 ±0.02 |
| 4000 | 70.69 ±0.09 | 70.08 ±0.26 | 10.66 ±0.05 |

(d) **CIFAR-100** with the **MIR** framework (AAA).

| $|\mathcal{M}|$ | ER (vanilla) Model Based | MERS ProbCover | | | MERS MaxHerding | | |
|---|---|---|---|---|---|---|---|
| | | Model Based | SimCLR | MERS | Model Based | SimCLR | MERS |
| 100 | 38.09 ±0.20 | 40.41 ±0.22 | 39.81 ±0.16 | **40.67** ±0.14 | **40.79** ±0.12 | 40.42 ±0.36 | 40.09 ±0.29 |
| 300 | 44.25 ±0.22 | **46.80** ±0.13 | 46.11 ±0.36 | **46.78** ±0.28 | 45.41 ±0.37 | 46.48 ±0.19 | **46.88** ±0.11 |
| 500 | 47.89 ±0.32 | 49.95 ±0.20 | 49.98 ±0.07 | **50.15** ±0.21 | 49.78 ±0.50 | **50.37** ±0.28 | **50.13** ±0.26 |
| 1000 | 53.83 ±0.14 | 55.37 ±0.27 | 55.55 ±0.21 | 55.81 ±0.29 | 55.44 ±0.14 | 55.91 ±0.19 | **56.32** ±0.08 |
| 2000 | 60.52 ±0.17 | 61.55 ±0.18 | **61.64** ±0.26 | 61.57 ±0.23 | **61.70** ±0.14 | 61.60 ±0.07 | **61.67** ±0.06 |
| 4000 | 66.84 ±0.21 | **67.60** ±0.12 | 67.44 ±0.24 | **67.68** ±0.21 | **67.57** ±0.15 | **67.50** ±0.11 | 67.12 ±0.19 |
| 5000 | 68.41 ±0.18 | **69.09** ±0.21 | 68.93 ±0.17 | **68.96** ±0.15 | 68.76 ±0.20 | 68.81 ±0.11 | 68.86 ±0.11 |

Table 10: **Average Accumulated Accuracy (AAA)**, averaged over 5 independent runs (mean $\pm$ standard error). For each buffer size, the best aaa is in bold; results within the standard error of the best are also bolded.

(a) **TinyImageNet** with the **ER** framework (AAA).

| $|\mathcal{M}|$ | ER (vanilla) Model Based | MERS ProbCover | | | MERS MaxHerding | | |
|---|---|---|---|---|---|---|---|
| | | Model Based | SimCLR | *MERS* | Model Based | SimCLR | *MERS* |
| 200 | 20.97 $\pm$0.09 | **21.34** $\pm$0.07 | 21.16 $\pm$0.04 | 21.10 $\pm$0.04 | 21.13 $\pm$0.12 | 21.12 $\pm$0.04 | 21.14 $\pm$0.07 |
| 400 | 20.91 $\pm$0.13 | 21.44 $\pm$0.08 | 21.18 $\pm$0.09 | **21.57** $\pm$0.04 | **21.55** $\pm$0.05 | 21.32 $\pm$0.08 | 21.28 $\pm$0.09 |
| 600 | 21.03 $\pm$0.07 | **22.11** $\pm$0.11 | 21.47 $\pm$0.06 | 21.98 $\pm$0.08 | 21.96 $\pm$0.08 | 21.62 $\pm$0.05 | 21.78 $\pm$0.08 |
| 1000 | 22.17 $\pm$0.10 | **23.32** $\pm$0.16 | 22.55 $\pm$0.10 | **23.36** $\pm$0.12 | **23.24** $\pm$0.05 | 22.99 $\pm$0.10 | 22.91 $\pm$0.08 |
| 2000 | 25.38 $\pm$0.09 | 27.44 $\pm$0.06 | 26.62 $\pm$0.09 | **28.12** $\pm$0.19 | 27.59 $\pm$0.07 | 26.36 $\pm$0.05 | 26.78 $\pm$0.15 |
| 4000 | 33.17 $\pm$0.29 | **35.66** $\pm$0.29 | 34.52 $\pm$0.12 | **35.77** $\pm$0.20 | 35.22 $\pm$0.20 | 33.62 $\pm$0.24 | 34.17 $\pm$0.24 |
| 6000 | 40.14 $\pm$0.22 | **41.61** $\pm$0.20 | 40.53 $\pm$0.26 | 41.41 $\pm$0.07 | 40.23 $\pm$0.23 | 39.82 $\pm$0.21 | 39.97 $\pm$0.23 |

(b) **TinyImageNet** with the **ER-ACE** framework (AAA).

| $|\mathcal{M}|$ | Herding Model Based | TEAL Model Based |
|---|---|---|
| 200 | 28.46 $\pm$0.12 | 29.41 $\pm$0.16 |
| 400 | 29.96 $\pm$0.22 | 31.30 $\pm$0.08 |
| 600 | 31.07 $\pm$0.18 | 32.53 $\pm$0.11 |
| 1000 | 32.54 $\pm$0.23 | 34.53 $\pm$0.08 |
| 2000 | 35.28 $\pm$0.20 | 37.59 $\pm$0.19 |
| 4000 | 39.15 $\pm$0.12 | 41.29 $\pm$0.13 |
| 6000 | 42.14 $\pm$0.19 | 43.38 $\pm$0.24 |

(c) **CIFAR-100** with the **MIR** framework (AAA).

| $|\mathcal{M}|$ | Herding Model Based | RM Model Based |
|---|---|---|
| 100 | 39.23 $\pm$0.47 | 37.69 $\pm$0.23 |
| 300 | 45.39 $\pm$0.16 | 43.88 $\pm$0.23 |
| 500 | 48.74 $\pm$0.19 | 48.04 $\pm$0.21 |
| 1000 | 55.64 $\pm$0.28 | 54.31 $\pm$0.14 |
| 2000 | 61.74 $\pm$0.21 | 60.66 $\pm$0.28 |
| 4000 | 67.94 $\pm$0.23 | 66.17 $\pm$0.08 |
| 5000 | 69.56 $\pm$0.21 | 68.22 $\pm$0.06 |

Table 11: **Final Averaged Accuracy (FAA)**, averaged over 5 independent runs (mean $\pm$ standard error). For each buffer size, the best AAA is in bold; results within the standard error of the best are also bolded.

(a) **CIFAR-100** with the **ER-ACE** framework.

| | | *MERS MaxHerding* | | |
|---|---|---|---|---|
| $|\mathcal{M}|$ | VICReg | VICReg + Model Based | DINOv2 | DINOv2 + Model Based |
| 100 | 25.33 $\pm0.39$ | 28.96 $\pm0.31$ | 27.96 $\pm0.22$ | **29.80** $\pm0.19$ |
| 300 | 37.62 $\pm0.49$ | **39.61** $\pm0.21$ | 38.01 $\pm0.20$ | **39.52** $\pm0.15$ |
| 500 | 41.93 $\pm0.40$ | **44.23** $\pm0.27$ | 42.68 $\pm0.13$ | **43.95** $\pm0.33$ |
| 1000 | 48.52 $\pm0.25$ | **50.03** $\pm0.27$ | 48.40 $\pm0.10$ | **49.86** $\pm0.08$ |
| 2000 | 53.93 $\pm0.18$ | 54.82 $\pm0.17$ | 54.27 $\pm0.18$ | **55.13** $\pm0.09$ |
| 4000 | **59.77** $\pm0.31$ | **59.82** $\pm0.18$ | 59.44 $\pm0.12$ | 59.44 $\pm0.19$ |
| 5000 | 60.82 $\pm0.25$ | **60.84** $\pm0.19$ | 60.30 $\pm0.27$ | **60.73** $\pm0.19$ |

(b) **CIFAR-100** with the **ER-ACE** framework (FAA).

| | | *MERS MaxHerding* | | |
|---|---|---|---|---|
| $|\mathcal{M}|$ | VICReg | VICReg + Model Based | DINOv2 | DINOv2 + Model Based |
| 100 | 25.33 $\pm0.39$ | 28.96 $\pm0.31$ | 27.96 $\pm0.22$ | **29.80** $\pm0.19$ |
| 300 | 37.62 $\pm0.49$ | **39.61** $\pm0.21$ | 38.01 $\pm0.20$ | **39.52** $\pm0.15$ |
| 500 | 41.93 $\pm0.40$ | **44.23** $\pm0.27$ | 42.68 $\pm0.13$ | **43.95** $\pm0.33$ |
| 1000 | 48.52 $\pm0.25$ | **50.03** $\pm0.27$ | 48.40 $\pm0.10$ | **49.86** $\pm0.08$ |
| 2000 | 53.93 $\pm0.18$ | 54.82 $\pm0.17$ | 54.27 $\pm0.18$ | **55.13** $\pm0.09$ |
| 4000 | **59.77** $\pm0.31$ | **59.82** $\pm0.18$ | 59.44 $\pm0.12$ | 59.44 $\pm0.19$ |
| 5000 | 60.82 $\pm0.25$ | **60.84** $\pm0.19$ | 60.30 $\pm0.27$ | **60.73** $\pm0.19$ |

Table 12: **Average Accumulated Accuracy (AAA)**, averaged over 5 independent runs (mean $\pm$ standard error). For each buffer size, the best AAA is in bold; results within the standard error of the best are also bolded.

(a) **CIFAR-100** with the **ER-ACE** framework (AAA).

| | | *MERS MaxHerding* | | |
|---|---|---|---|---|
| $|\mathcal{M}|$ | VICReg | VICReg + Model Based | DINOv2 | DINOv2 + Model Based |
| 100 | 45.07 $\pm0.27$ | **47.02** $\pm0.15$ | 45.75 $\pm0.17$ | 46.54 $\pm0.11$ |
| 300 | 53.70 $\pm0.29$ | **55.25** $\pm0.13$ | 54.58 $\pm0.16$ | 55.01 $\pm0.24$ |
| 500 | 57.84 $\pm0.36$ | **58.54** $\pm0.32$ | 58.13 $\pm0.11$ | **58.54** $\pm0.18$ |
| 1000 | 62.50 $\pm0.16$ | **63.15** $\pm0.33$ | 62.33 $\pm0.31$ | **62.86** $\pm0.22$ |
| 2000 | 66.41 $\pm0.19$ | **66.99** $\pm0.10$ | 66.55 $\pm0.25$ | **67.13** $\pm0.23$ |
| 4000 | **70.62** $\pm0.26$ | 70.47 $\pm0.14$ | **70.52** $\pm0.07$ | 69.94 $\pm0.17$ |
| 5000 | **71.56** $\pm0.37$ | 71.22 $\pm0.09$ | 71.11 $\pm0.20$ | **71.42** $\pm0.20$ |

(b) **CIFAR-100** with the **ER-ACE** framework (AAA).

| | | *MERS ProbCover* | | |
|---|---|---|---|---|
| $|\mathcal{M}|$ | VICReg | VICReg + Model Based | DINOv2 | DINOv2 + Model Based |
| 100 | 44.55 $\pm0.17$ | **45.95** $\pm0.25$ | 44.51 $\pm0.10$ | **45.74** $\pm0.22$ |
| 300 | 53.71 $\pm0.48$ | **54.72** $\pm0.22$ | 53.89 $\pm0.18$ | **54.64** $\pm0.19$ |
| 500 | 57.22 $\pm0.21$ | **58.62** $\pm0.40$ | 57.67 $\pm0.19$ | **58.51** $\pm0.15$ |
| 1000 | 62.58 $\pm0.20$ | **63.35** $\pm0.39$ | 61.61 $\pm0.14$ | **62.97** $\pm0.15$ |
| 4000 | 70.37 $\pm0.17$ | **71.10** $\pm0.05$ | 70.63 $\pm0.22$ | 70.73 $\pm0.19$ |
| 5000 | **71.90** $\pm0.29$ | **71.83** $\pm0.36$ | 71.61 $\pm0.15$ | 71.60 $\pm0.20$ |

Table 13: **FAA** for the class order defined by **seed 42**, averaged over 5 independent runs (mean ± standard error). Several sample-selection strategies and embedding spaces are compared across multiple replay-buffer sizes ($|\mathcal{M}|$).

(a) FAA on CIFAR-100 **ER ACE**.

| Buffer | ER (vanilla) Model Based | *MERS ProbCover* Model Based | SimCLR | *MERS* | *MERS MaxHerding* Model Based | SimCLR | *MERS* |
|---|---|---|---|---|---|---|---|
| 100 | $20.79_{\pm0.27}$ | $29.81_{\pm0.20}$ | $27.82_{\pm0.25}$ | $29.35_{\pm0.24}$ | $29.42_{\pm0.11}$ | $29.20_{\pm0.33}$ | $29.89_{\pm0.22}$ |
| 300 | $31.76_{\pm0.07}$ | $38.84_{\pm0.19}$ | $37.78_{\pm0.14}$ | $39.47_{\pm0.28}$ | $38.16_{\pm0.35}$ | $38.73_{\pm0.36}$ | $39.60_{\pm0.25}$ |
| 500 | $35.80_{\pm0.32}$ | $42.46_{\pm0.23}$ | $42.25_{\pm0.19}$ | $43.28_{\pm0.23}$ | $42.72_{\pm0.23}$ | $42.82_{\pm0.21}$ | $43.71_{\pm0.21}$ |
| 1000 | $42.27_{\pm0.22}$ | $47.69_{\pm0.23}$ | $48.11_{\pm0.19}$ | $48.98_{\pm0.27}$ | $47.77_{\pm0.21}$ | $48.89_{\pm0.27}$ | $50.00_{\pm0.16}$ |
| 2000 | $49.41_{\pm0.18}$ | $52.99_{\pm0.07}$ | $53.53_{\pm0.24}$ | $54.17_{\pm0.19}$ | $53.18_{\pm0.20}$ | $54.20_{\pm0.28}$ | $54.80_{\pm0.19}$ |
| 4000 | $55.32_{\pm0.24}$ | $58.03_{\pm0.11}$ | $58.79_{\pm0.24}$ | $59.28_{\pm0.18}$ | $58.52_{\pm0.20}$ | $58.56_{\pm0.34}$ | $59.03_{\pm0.19}$ |
| 5000 | $57.96_{\pm0.21}$ | $60.10_{\pm0.11}$ | $60.79_{\pm0.19}$ | $60.84_{\pm0.27}$ | $59.85_{\pm0.18}$ | $59.90_{\pm0.11}$ | $60.07_{\pm0.13}$ |

(b) FAA on CIFAR-100 **ER**.

| Buffer | ER (vanilla) Model Based | *MERS ProbCover* Model Based | SimCLR | *MERS* | *MERS MaxHerding* Model Based | SimCLR | *MERS* |
|---|---|---|---|---|---|---|---|
| 100 | $10.50_{\pm0.14}$ | $13.02_{\pm0.11}$ | $11.44_{\pm0.13}$ | $12.18_{\pm0.08}$ | $12.53_{\pm0.18}$ | $12.24_{\pm0.07}$ | $12.79_{\pm0.12}$ |
| 300 | $14.67_{\pm0.24}$ | $20.32_{\pm0.26}$ | $19.01_{\pm0.34}$ | $20.33_{\pm0.21}$ | $19.62_{\pm0.17}$ | $18.83_{\pm0.56}$ | $19.52_{\pm0.33}$ |
| 500 | $19.86_{\pm0.31}$ | $25.37_{\pm0.18}$ | $23.68_{\pm0.35}$ | $25.01_{\pm0.44}$ | $25.73_{\pm0.22}$ | $24.25_{\pm0.22}$ | $25.74_{\pm0.55}$ |
| 1000 | $28.48_{\pm0.22}$ | $34.37_{\pm0.33}$ | $33.62_{\pm0.29}$ | $35.18_{\pm0.21}$ | $34.54_{\pm0.34}$ | $34.43_{\pm0.35}$ | $35.40_{\pm0.20}$ |
| 2000 | $40.45_{\pm0.23}$ | $43.84_{\pm0.32}$ | $44.34_{\pm0.31}$ | $45.38_{\pm0.28}$ | $44.62_{\pm0.19}$ | $44.76_{\pm0.37}$ | $45.58_{\pm0.40}$ |
| 4000 | $51.23_{\pm0.22}$ | $53.91_{\pm0.20}$ | $54.37_{\pm0.20}$ | $54.97_{\pm0.27}$ | $54.69_{\pm0.32}$ | $54.01_{\pm0.16}$ | $54.81_{\pm0.16}$ |
| 5000 | $55.03_{\pm0.20}$ | $56.75_{\pm0.13}$ | $57.16_{\pm0.25}$ | $57.79_{\pm0.22}$ | $56.67_{\pm0.23}$ | $56.49_{\pm0.21}$ | $57.06_{\pm0.23}$ |

(c) FAA on TinyImageNet **ER ACE**.

| Buffer | ER (vanilla) Model Based | *MERS ProbCover* Model Based | SimCLR | *MERS* | *MERS MaxHerding* Model Based | SimCLR | *MERS* |
|---|---|---|---|---|---|---|---|
| 200 | $11.89_{\pm0.13}$ | $13.95_{\pm0.17}$ | $12.58_{\pm0.05}$ | $13.33_{\pm0.09}$ | $13.54_{\pm0.07}$ | $13.50_{\pm0.14}$ | $13.91_{\pm0.25}$ |
| 400 | $13.27_{\pm0.12}$ | $15.69_{\pm0.12}$ | $14.33_{\pm0.12}$ | $15.16_{\pm0.11}$ | $14.72_{\pm0.21}$ | $15.00_{\pm0.10}$ | $15.35_{\pm0.19}$ |
| 600 | $13.47_{\pm0.08}$ | $16.44_{\pm0.06}$ | $15.42_{\pm0.19}$ | $16.64_{\pm0.24}$ | $15.71_{\pm0.18}$ | $16.07_{\pm0.10}$ | $16.37_{\pm0.31}$ |
| 1000 | $14.50_{\pm0.16}$ | $18.16_{\pm0.19}$ | $16.99_{\pm0.19}$ | $18.44_{\pm0.11}$ | $17.51_{\pm0.16}$ | $17.26_{\pm0.15}$ | $17.89_{\pm0.12}$ |
| 2000 | $16.59_{\pm0.15}$ | $20.50_{\pm0.21}$ | $19.71_{\pm0.19}$ | $21.03_{\pm0.09}$ | $20.08_{\pm0.23}$ | $19.50_{\pm0.20}$ | $20.26_{\pm0.27}$ |
| 4000 | $19.11_{\pm0.13}$ | $23.09_{\pm0.15}$ | $22.94_{\pm0.17}$ | $24.45_{\pm0.20}$ | $23.18_{\pm0.21}$ | $22.43_{\pm0.33}$ | $22.94_{\pm0.20}$ |
| 6000 | $22.57_{\pm0.06}$ | $25.41_{\pm0.20}$ | $25.42_{\pm0.30}$ | $26.40_{\pm0.26}$ | $25.66_{\pm0.19}$ | $24.66_{\pm0.18}$ | $25.02_{\pm0.15}$ |

Table 14: **FAA** for the class order defined by **seed 35**, averaged over 5 independent runs (mean ± standard error). Several sample-selection strategies and embedding spaces are compared across multiple replay-buffer sizes ($|\mathcal{M}|$).

(a) FAA on CIFAR-100 **ER ACE**.

| Buffer | ER (vanilla) Model Based | MERS ProbCover Model Based | SimCLR | MERS | MERS MaxHerding Model Based | SimCLR | MERS |
|---|---|---|---|---|---|---|---|
| 100 | 20.59 $\pm 0.23$ | 27.64 $\pm 0.44$ | 25.67 $\pm 0.45$ | 27.42 $\pm 0.32$ | 28.10 $\pm 0.30$ | 28.30 $\pm 0.44$ | 29.35 $\pm 0.25$ |
| 300 | 28.61 $\pm 0.05$ | 37.84 $\pm 0.14$ | 36.90 $\pm 0.31$ | 38.88 $\pm 0.21$ | 37.70 $\pm 0.34$ | 37.63 $\pm 0.31$ | 39.19 $\pm 0.19$ |
| 500 | 35.30 $\pm 0.19$ | 42.23 $\pm 0.19$ | 41.75 $\pm 0.18$ | 43.55 $\pm 0.23$ | 42.02 $\pm 0.25$ | 42.42 $\pm 0.13$ | 44.02 $\pm 0.38$ |
| 1000 | 41.99 $\pm 0.16$ | 48.18 $\pm 0.22$ | 48.36 $\pm 0.15$ | 48.96 $\pm 0.25$ | 47.89 $\pm 0.23$ | 48.71 $\pm 0.30$ | 49.63 $\pm 0.30$ |
| 2000 | 49.00 $\pm 0.23$ | 53.20 $\pm 0.22$ | 53.51 $\pm 0.08$ | 54.44 $\pm 0.28$ | 53.67 $\pm 0.22$ | 54.30 $\pm 0.14$ | 55.11 $\pm 0.10$ |
| 4000 | 56.89 $\pm 0.11$ | 59.01 $\pm 0.27$ | 59.18 $\pm 0.12$ | 59.73 $\pm 0.15$ | 59.30 $\pm 0.05$ | 59.23 $\pm 0.14$ | 59.67 $\pm 0.10$ |
| 5000 | 58.75 $\pm 0.22$ | 60.45 $\pm 0.18$ | 60.65 $\pm 0.11$ | 61.40 $\pm 0.22$ | 60.17 $\pm 0.11$ | 60.37 $\pm 0.09$ | 60.94 $\pm 0.07$ |

(b) FAA on CIFAR-100 **ER**.

| Buffer | ER (vanilla) Model Based | MERS ProbCover Model Based | SimCLR | MERS | MERS MaxHerding Model Based | SimCLR | MERS |
|---|---|---|---|---|---|---|---|
| 100 | 9.95 $\pm 0.07$ | 11.45 $\pm 0.11$ | 10.32 $\pm 0.06$ | 11.17 $\pm 0.17$ | 11.19 $\pm 0.05$ | 11.05 $\pm 0.18$ | 11.51 $\pm 0.01$ |
| 300 | 13.71 $\pm 0.09$ | 18.87 $\pm 0.11$ | 17.16 $\pm 0.14$ | 18.71 $\pm 0.25$ | 18.10 $\pm 0.34$ | 18.01 $\pm 0.22$ | 18.71 $\pm 0.25$ |
| 500 | 17.41 $\pm 0.38$ | 23.75 $\pm 0.39$ | 22.37 $\pm 0.34$ | 24.66 $\pm 0.14$ | 24.11 $\pm 0.15$ | 23.40 $\pm 0.15$ | 24.66 $\pm 0.29$ |
| 1000 | 27.44 $\pm 0.48$ | 33.50 $\pm 0.17$ | 32.51 $\pm 0.48$ | 33.99 $\pm 0.27$ | 33.70 $\pm 0.20$ | 33.27 $\pm 0.16$ | 34.64 $\pm 0.31$ |
| 2000 | 39.78 $\pm 0.30$ | 43.73 $\pm 0.01$ | 43.74 $\pm 0.30$ | 44.02 $\pm 0.20$ | 44.06 $\pm 0.30$ | 44.01 $\pm 0.20$ | 45.22 $\pm 0.16$ |

(c) FAA on TinyImagenet **ER ACE**.

| Buffer | ER (vanilla) Model Based | MERS ProbCover Model Based | SimCLR | MERS | MERS MaxHerding Model Based | SimCLR | MERS |
|---|---|---|---|---|---|---|---|
| 200 | 11.39 $\pm 0.10$ | 13.23 $\pm 0.12$ | 12.22 $\pm 0.14$ | 12.75 $\pm 0.13$ | 13.11 $\pm 0.08$ | 12.71 $\pm 0.11$ | 12.95 $\pm 0.09$ |
| 400 | 11.98 $\pm 0.24$ | 15.09 $\pm 0.21$ | 13.70 $\pm 0.16$ | 14.71 $\pm 0.24$ | 13.84 $\pm 0.15$ | 14.12 $\pm 0.21$ | 14.51 $\pm 0.16$ |
| 600 | 12.90 $\pm 0.13$ | 16.18 $\pm 0.12$ | 14.68 $\pm 0.08$ | 15.78 $\pm 0.18$ | 14.97 $\pm 0.19$ | 15.48 $\pm 0.13$ | 15.14 $\pm 0.06$ |
| 1000 | 14.14 $\pm 0.09$ | 17.67 $\pm 0.26$ | 16.21 $\pm 0.22$ | 17.47 $\pm 0.15$ | 16.61 $\pm 0.11$ | 16.32 $\pm 0.18$ | 16.77 $\pm 0.10$ |
| 2000 | 15.94 $\pm 0.16$ | 19.88 $\pm 0.24$ | 18.60 $\pm 0.23$ | 20.42 $\pm 0.29$ | 19.70 $\pm 0.34$ | 19.01 $\pm 0.14$ | 19.21 $\pm 0.22$ |
| 4000 | 19.42 $\pm 0.22$ | 22.86 $\pm 0.13$ | 23.05 $\pm 0.35$ | 24.08 $\pm 0.07$ | 22.80 $\pm 0.12$ | 21.84 $\pm 0.28$ | 21.84 $\pm 0.23$ |
| 6000 | 22.05 $\pm 0.25$ | 25.98 $\pm 0.34$ | 25.63 $\pm 0.30$ | 26.53 $\pm 0.13$ | 25.14 $\pm 0.23$ | 24.43 $\pm 0.28$ | 25.23 $\pm 0.25$ |

Table 15: **AAA** for the class order defined by **seed 42**, averaged over 5 independent runs (mean ± standard error). Several sample-selection strategies and embedding spaces are compared across multiple replay-buffer sizes ($|\mathcal{M}|$).

(a) AAA on CIFAR-100 **ER ACE**.

| Buffer | ER (vanilla) Model Based | *MERS ProbCover* Model Based | SimCLR | *MERS* | *MERS MaxHerding* Model Based | SimCLR | *MERS* |
|---|---|---|---|---|---|---|---|
| 100 | 39.94 ±0.09 | 46.09 ±0.08 | 45.60 ±0.09 | 46.90 ±0.18 | 46.21 ±0.18 | 46.17 ±0.11 | 46.78 ±0.27 |
| 300 | 49.19 ±0.10 | 53.33 ±0.07 | 53.62 ±0.09 | 54.30 ±0.13 | 53.46 ±0.30 | 53.92 ±0.19 | 54.68 ±0.14 |
| 500 | 52.85 ±0.08 | 56.55 ±0.15 | 56.76 ±0.12 | 57.07 ±0.26 | 56.52 ±0.12 | 57.34 ±0.10 | 57.77 ±0.07 |
| 1000 | 57.60 ±0.13 | 60.65 ±0.09 | 60.90 ±0.10 | 61.46 ±0.06 | 60.60 ±0.23 | 61.25 ±0.06 | 61.97 ±0.17 |
| 2000 | 62.35 ±0.14 | 64.36 ±0.20 | 64.85 ±0.11 | 64.91 ±0.12 | 64.29 ±0.13 | 65.01 ±0.08 | 65.26 ±0.15 |
| 4000 | 66.73 ±0.17 | 68.22 ±0.09 | 68.39 ±0.17 | 68.67 ±0.14 | 68.20 ±0.14 | 67.97 ±0.08 | 68.16 ±0.07 |
| 5000 | 68.32 ±0.16 | 69.36 ±0.08 | 69.92 ±0.15 | 69.80 ±0.17 | 69.40 ±0.14 | 69.07 ±0.04 | 69.20 ±0.10 |

(b) AAA on CIFAR-100 **ER**.

| Buffer | ER (vanilla) Model Based | *MERS ProbCover* Model Based | SimCLR | *MERS* | *MERS MaxHerding* Model Based | SimCLR | *MERS* |
|---|---|---|---|---|---|---|---|
| 100 | 28.19 ±0.11 | 30.89 ±0.07 | 29.72 ±0.24 | 30.51 ±0.16 | 30.31 ±0.11 | 30.37 ±0.15 | 30.49 ±0.15 |
| 300 | 34.42 ±0.42 | 38.55 ±0.37 | 38.12 ±0.33 | 39.30 ±0.13 | 38.44 ±0.25 | 37.92 ±0.46 | 38.37 ±0.42 |
| 500 | 40.31 ±0.21 | 43.90 ±0.27 | 42.60 ±0.39 | 43.55 ±0.34 | 43.74 ±0.09 | 43.58 ±0.37 | 44.68 ±0.31 |
| 1000 | 48.62 ±0.24 | 51.78 ±0.20 | 51.86 ±0.22 | 52.58 ±0.35 | 51.67 ±0.37 | 52.27 ±0.31 | 52.71 ±0.25 |
| 2000 | 58.66 ±0.23 | 59.70 ±0.47 | 60.57 ±0.20 | 61.48 ±0.18 | 60.68 ±0.13 | 60.73 ±0.37 | 60.49 ±0.28 |
| 4000 | 66.49 ±0.18 | 67.67 ±0.23 | 67.41 ±0.17 | 68.30 ±0.32 | 68.10 ±0.12 | 67.35 ±0.27 | 68.12 ±0.11 |
| 5000 | 68.89 ±0.17 | 69.58 ±0.18 | 69.53 ±0.22 | 70.13 ±0.21 | 69.02 ±0.21 | 69.21 ±0.32 | 69.43 ±0.05 |

(c) AAA on TinyImageNet **ER ACE**.

| Buffer | ER (vanilla) Model Based | *MERS ProbCover* Model Based | SimCLR | *MERS* | *MERS MaxHerding* Model Based | SimCLR | *MERS* |
|---|---|---|---|---|---|---|---|
| 200 | 25.92 ±0.07 | 28.32 ±0.06 | 27.43 ±0.10 | 28.17 ±0.11 | 27.91 ±0.09 | 27.93 ±0.09 | 28.20 ±0.08 |
| 400 | 27.78 ±0.16 | 30.60 ±0.13 | 29.61 ±0.07 | 30.50 ±0.09 | 29.73 ±0.05 | 29.94 ±0.10 | 30.10 ±0.10 |
| 600 | 28.96 ±0.07 | 31.60 ±0.13 | 30.94 ±0.09 | 31.82 ±0.08 | 31.18 ±0.09 | 31.40 ±0.15 | 31.56 ±0.13 |
| 1000 | 30.49 ±0.05 | 33.60 ±0.15 | 33.05 ±0.13 | 34.08 ±0.10 | 33.43 ±0.12 | 33.12 ±0.20 | 33.22 ±0.11 |
| 2000 | 33.23 ±0.13 | 36.09 ±0.13 | 35.84 ±0.11 | 36.86 ±0.05 | 36.08 ±0.14 | 35.51 ±0.14 | 36.04 ±0.20 |
| 4000 | 36.95 ±0.13 | 39.32 ±0.13 | 39.06 ±0.10 | 39.87 ±0.12 | 39.10 ±0.12 | 38.47 ±0.09 | 38.57 ±0.12 |
| 6000 | 39.66 ±0.12 | 40.90 ±0.12 | 41.19 ±0.10 | 41.67 ±0.08 | 40.94 ±0.15 | 40.08 ±0.10 | 40.26 ±0.16 |

Table 16: **AAA** for the class order defined by **seed 35**, averaged over 5 independent runs (mean ± standard error). Several sample-selection strategies and embedding spaces are compared across multiple replay-buffer sizes ($|\mathcal{M}|$).

(a) AAA on CIFAR-100 **ER ACE**.

| Buffer | ER (vanilla) Model Based | MERS ProbCover Model Based | SimCLR | MERS | MERS MaxHerding Model Based | SimCLR | MERS |
|---|---|---|---|---|---|---|---|
| 100 | 41.79 $\pm 0.14$ | 47.76 $\pm 0.18$ | 47.34 $\pm 0.14$ | 48.44 $\pm 0.08$ | 48.51 $\pm 0.17$ | 48.63 $\pm 0.15$ | 49.18 $\pm 0.11$ |
| 300 | 51.26 $\pm 0.11$ | 56.10 $\pm 0.12$ | 56.54 $\pm 0.18$ | 57.50 $\pm 0.13$ | 56.33 $\pm 0.17$ | 57.15 $\pm 0.14$ | 57.78 $\pm 0.13$ |
| 500 | 56.12 $\pm 0.28$ | 59.79 $\pm 0.14$ | 60.32 $\pm 0.15$ | 61.35 $\pm 0.08$ | 60.28 $\pm 0.12$ | 60.63 $\pm 0.12$ | 61.45 $\pm 0.11$ |
| 1000 | 61.84 $\pm 0.20$ | 64.49 $\pm 0.20$ | 65.41 $\pm 0.12$ | 65.54 $\pm 0.14$ | 64.56 $\pm 0.12$ | 65.04 $\pm 0.17$ | 65.79 $\pm 0.22$ |
| 2000 | 66.46 $\pm 0.05$ | 68.70 $\pm 0.17$ | 68.92 $\pm 0.18$ | 69.02 $\pm 0.15$ | 68.87 $\pm 0.12$ | 69.22 $\pm 0.09$ | 69.29 $\pm 0.17$ |
| 4000 | 71.57 $\pm 0.10$ | 72.34 $\pm 0.17$ | 72.52 $\pm 0.11$ | 73.00 $\pm 0.03$ | 72.53 $\pm 0.06$ | 72.30 $\pm 0.21$ | 72.44 $\pm 0.13$ |
| 5000 | 72.95 $\pm 0.09$ | 73.61 $\pm 0.10$ | 73.48 $\pm 0.09$ | 74.22 $\pm 0.12$ | 73.14 $\pm 0.09$ | 73.36 $\pm 0.14$ | 73.66 $\pm 0.03$ |

(b) AAA on CIFAR-100 **ER**.

| Buffer | ER (vanilla) Model Based | MERS ProbCover Model Based | SimCLR | MERS | MERS MaxHerding Model Based | SimCLR | MERS |
|---|---|---|---|---|---|---|---|
| 100 | 29.82 $\pm 0.13$ | 32.42 $\pm 0.04$ | 31.62 $\pm 0.12$ | 32.58 $\pm 0.20$ | 32.28 $\pm 0.08$ | 32.09 $\pm 0.16$ | 32.58 $\pm 0.17$ |
| 300 | 37.89 $\pm 0.08$ | 42.18 $\pm 0.13$ | 41.20 $\pm 0.14$ | 42.32 $\pm 0.15$ | 41.33 $\pm 0.18$ | 41.74 $\pm 0.14$ | 42.39 $\pm 0.04$ |
| 500 | 43.07 $\pm 0.17$ | 47.15 $\pm 0.13$ | 47.20 $\pm 0.09$ | 48.48 $\pm 0.09$ | 47.70 $\pm 0.09$ | 47.64 $\pm 0.12$ | 48.52 $\pm 0.19$ |
| 1000 | 52.56 $\pm 0.13$ | 55.93 $\pm 0.05$ | 55.92 $\pm 0.29$ | 56.60 $\pm 0.31$ | 56.30 $\pm 0.08$ | 56.35 $\pm 0.20$ | 57.20 $\pm 0.11$ |
| 2000 | 62.59 $\pm 0.15$ | 64.42 $\pm 0.21$ | 64.67 $\pm 0.11$ | 64.56 $\pm 0.04$ | 64.45 $\pm 0.13$ | 64.52 $\pm 0.07$ | 64.96 $\pm 0.10$ |

(c) AAA on TinyImageNet **ER-ACE**.

| Buffer | ER (vanilla) Model Based | MERS ProbCover Model Based | SimCLR | MERS | MERS MaxHerding Model Based | SimCLR | MERS |
|---|---|---|---|---|---|---|---|
| 200 | 26.65 $\pm 0.04$ | 28.49 $\pm 0.11$ | 27.57 $\pm 0.06$ | 28.24 $\pm 0.09$ | 28.16 $\pm 0.19$ | 28.19 $\pm 0.05$ | 28.31 $\pm 0.12$ |
| 400 | 28.01 $\pm 0.07$ | 30.93 $\pm 0.23$ | 29.82 $\pm 0.13$ | 30.79 $\pm 0.15$ | 30.06 $\pm 0.02$ | 30.28 $\pm 0.12$ | 30.49 $\pm 0.12$ |
| 600 | 29.02 $\pm 0.12$ | 32.01 $\pm 0.09$ | 31.05 $\pm 0.16$ | 32.21 $\pm 0.14$ | 31.76 $\pm 0.18$ | 31.45 $\pm 0.09$ | 31.66 $\pm 0.08$ |
| 1000 | 31.03 $\pm 0.15$ | 33.92 $\pm 0.13$ | 32.97 $\pm 0.07$ | 34.43 $\pm 0.16$ | 33.52 $\pm 0.14$ | 33.15 $\pm 0.16$ | 33.11 $\pm 0.12$ |
| 2000 | 34.01 $\pm 0.16$ | 36.25 $\pm 0.22$ | 35.97 $\pm 0.17$ | 36.96 $\pm 0.10$ | 36.65 $\pm 0.11$ | 36.11 $\pm 0.18$ | 36.11 $\pm 0.19$ |
| 4000 | 37.83 $\pm 0.13$ | 39.51 $\pm 0.11$ | 39.78 $\pm 0.21$ | 40.28 $\pm 0.12$ | 39.37 $\pm 0.13$ | 38.70 $\pm 0.10$ | 39.05 $\pm 0.11$ |
| 6000 | 40.15 $\pm 0.24$ | 42.05 $\pm 0.26$ | 41.66 $\pm 0.15$ | 42.57 $\pm 0.15$ | 41.37 $\pm 0.14$ | 40.76 $\pm 0.24$ | 41.43 $\pm 0.04$ |

(d) AAA on TinyImageNet **ER**.

| Buffer | ER (vanilla) Model Based | MERS ProbCover Model Based | SimCLR | MERS | MERS MaxHerding Model Based | SimCLR | MERS |
|---|---|---|---|---|---|---|---|
| 200 | 21.09 $\pm 0.10$ | 21.06 $\pm 0.04$ | 21.03 $\pm 0.02$ | 21.00 $\pm 0.09$ | 21.12 $\pm 0.13$ | 21.22 $\pm 0.02$ | 21.16 $\pm 0.12$ |
| 400 | 20.94 $\pm 0.07$ | 21.48 $\pm 0.11$ | 21.06 $\pm 0.04$ | 21.59 $\pm 0.06$ | 21.56 $\pm 0.05$ | 21.34 $\pm 0.05$ | 21.33 $\pm 0.09$ |
| 600 | 21.16 $\pm 0.10$ | 22.15 $\pm 0.09$ | 21.59 $\pm 0.09$ | 21.91 $\pm 0.11$ | 22.17 $\pm 0.10$ | 21.75 $\pm 0.08$ | 21.78 $\pm 0.06$ |
| 1000 | 21.91 $\pm 0.15$ | 23.30 $\pm 0.05$ | 22.91 $\pm 0.15$ | 23.64 $\pm 0.12$ | 23.30 $\pm 0.10$ | 22.82 $\pm 0.09$ | 22.83 $\pm 0.08$ |
| 2000 | 25.57 $\pm 0.14$ | 27.72 $\pm 0.14$ | 26.72 $\pm 0.10$ | 27.64 $\pm 0.09$ | 27.25 $\pm 0.15$ | 26.46 $\pm 0.16$ | 27.03 $\pm 0.10$ |
| 4000 | 33.03 $\pm 0.11$ | 35.37 $\pm 0.30$ | 34.41 $\pm 0.18$ | 36.29 $\pm 0.15$ | 35.24 $\pm 0.17$ | 34.08 $\pm 0.08$ | 34.45 $\pm 0.23$ |
| 6000 | 39.88 $\pm 0.15$ | 41.56 $\pm 0.17$ | 41.02 $\pm 0.17$ | 41.70 $\pm 0.14$ | 40.87 $\pm 0.24$ | 39.76 $\pm 0.18$ | 40.65 $\pm 0.07$ |

