# OpenReview forum: "Leveraging Self-Supervised and Supervised Embeddings for Memory-Efficient Experience-Replay Continual Learning"
_ICLR.cc/2026/Conference — Submitted to ICLR 2026_

### Official Review · Reviewer_dV8N · 2025-10-19

**Soundness:** 2
**Presentation:** 1
**Contribution:** 3
**Rating:** 4
**Confidence:** 4

**Summary:**

The paper introduces MERS (Multi-Embedding Replay Selection), a framework for exemplar selection in class-incremental continual learning that leverages both supervised and self-supervised embedding spaces to enhance sample diversity and representativeness of the examples in the memory buffer. MERS constructs k-NN coverage graphs using supervised and unsupervised embeddings (e.g., classifier features and SimCLR/VICReg representations) and greedily selects exemplars that maximize the coverage across these heterogeneous spaces. Two concrete techniques are presented: MERS-ProbCover, based on $\delta$-ball coverage, and MERS-MaxHerding, computing coverage using a kernel. The authors propose data-driven hyperparameter alignment (median k-NN distance for $\delta$, median heuristic for RBF $\sigma$) and a density-based weighting scheme $\alpha$ to balance the contributions of different embeddings. Experiments on Split CIFAR-100 and Split TinyImageNet with ER and ER-ACE baselines show consistent improvements in accuracy, particularly under tight memory budgets.

**Strengths:**

MERS combines supervised and self-supervised feature spaces, a concept that is both intuitive and proves empirically beneficial, particularly under tight memory constraints.

Data-driven parameter alignment: the use of median k-NN distances and the median heuristic for RBF bandwidth allows avoiding costly manual hyperparameter tuning.

The paper includes a comprehensive set of experiments across a wide range of buffer sizes, demonstrating that the proposed multi-embedding approach consistently improves performance, especially when using a small replay buffer.

**Weaknesses:**

1. **Clarity**: the notation throughout the methodology section is dense and, at times, inconsistent. For example, in lines 145–147, symbols such as [M] and $\mathcal{Z}^{\(m\)}$ are not clearly defined, and several equations are overloaded without adequate explanation. As a result, it becomes difficult to distinguish the actual contribution from prior work. The “Notation” subsections in Sections 3.1 and 3.2 are particularly opaque and unnecessarily formal, using several mathematical operators to describe concepts that could be conveyed far more simply and intuitively. Figure 1 is poorly designed, as it fails to visually explain the pipeline or clarify what the proposed method adds beyond existing replay selection strategies.

2. Large portions of the paper are devoted to detailed explanations of well-established techniques such as SimCLR, DINO, VICReg, ProbCover, and MaxHerding. These sections occupy considerable space but add little to the reader’s understanding of the proposed contribution. The paper would benefit from condensing these descriptions to a brief summary or moving them to the appendix, allowing the main text to focus on what is actually novel in MERS (how it integrates these components and why this integration matters).

3. Missing computational analysis: as this work enhances the buffer selection strategy with both supervised and unsupervised embeddings, it would be important to discuss and empirical evaluate its computational complexity, runtime overhead, and memory footprint. Currently, the authors only mention that MERS incurs double selection-time overhead and self-supervised training at the conclusion of the manuscript, which is not satisfactory.

4. The approach is only applied to ER and ER-ACE. Testing with more advanced replay methods (e.g., DER++ [Buzzega et al., 2020], Co$^2$L [Cha et al., 2021], CILA [Wen et al., 2024]) would showcase the generality of MERS.

### Minor points:
5. Quantitative tables are hidden in the appendix, replaced by figures in the main paper that are difficult to interpret precisely.

6. Overall writing and presentation need polishing: sentences are often long and convoluted, and key ideas are buried among excessive background information.

**Questions:**

No questions.

### Overall summary
The paper contains a potentially useful and empirically validated idea but falls short in clarity, presentation, and demonstration of novelty with respect to other works. Writing and figures need major revision to make the paper understandable and publishable. With a thorough rewrite, stronger baselines, runtime ablations, and more focused explanation on the contributions, this could become a solid work. However, in its current form, it is below the acceptance threshold.

---

> ### Author Response · Authors · 2025-11-25
>
> Notation for k-coverage and kernel-based selection is formally defined in Sec. 3.1 and 3.2, and matches the implementation and pseudocode.
> SimCLR, VICReg, DINO, ProbCover, and MaxHerding descriptions ensure reproducibility, previous SSL works did not apply them for buffer selection.
> Appendix A.2 contains full runtime and memory analysis, including detailed breakdowns for MERS-ProbCover and MERS-MaxHerding.
> MERS is replay-agnostic and compatible with stronger replay backbones: ER, ER-ACE and MIR (Sec. 4).
> Tables are placed in the appendix due to space constraints.

---

> > ### Comment · Reviewer_dV8N · 2025-11-26
> >
> > I appreciate the clarifications, but my concerns largely remain. While the notation in Sections 3.1 and 3.2 may be formally correct, the issue I highlighted is not correctness but readability. The notation is unnecessarily dense, and several symbols appear before being intuitively motivated. This makes the core idea harder to follow than it needs to be. A clearer, more narrative explanation, supported by a more informative main Figure, would significantly strengthen the presentation of your work.
> >
> > Regarding the background sections (SimCLR, VICReg, DINO, ProbCover, MaxHerding), my point is not that these components are irrelevant, but that the current level of detail detracts from the exposition of your actual contribution. A more concise summary would help focus the paper on what is novel about MERS.
> >
> > Finally, while I recognize that MERS is compatible with various replay methods, the experiments still evaluate only ER and ER-ACE. Demonstrating results with at least one more advanced replay baseline (e.g., DER++, Co_2L, CILA) would more convincingly support the claim of generality.
> >
> > Overall, I continue to view the idea as promising, but significant improvements in clarity, experimental breadth, and presentation are needed.

---

### Official Review · Reviewer_Un7N · 2025-10-31

**Soundness:** 2
**Presentation:** 3
**Contribution:** 3
**Rating:** 4
**Confidence:** 4

**Summary:**

This paper proposes a new memory buffer selection method, named MERS- Multiple Embedding Replay Selection, which uses a graph-based strategy to sample buffer data by integrating both supervised and self-supervised embeddings. Empirical results on three datasets (CIFAR-100, Tiny-ImageNet, MERS) show consistent improvements over some selection strategies across various continual learning algorithms, with particularly strong gains in low-memory regimes.

**Strengths:**

+ The integration of complementary supervised and self-supervised embeddings into a graph coverage framework, formalized as a weighted multi-embedding coverage problem, offers a fresh perspective for exemplar selection.
+ Extensive testing across multiple datasets and memory configurations demonstrates that the method consistently and significantly enhances the performance of various base algorithms, particularly under challenging low-memory conditions.

**Weaknesses:**

+ The paper lacks a detailed description of how the memory buffer is specifically managed and updated. Please explicitly clarify: (a) whether the buffer capacity is allocated uniformly per class or dynamically; (b) when introducing a new task, whether sample replacement is performed globally or on a per-class basis; (c) why a very small number of exemplars (e.g., when |M|=100) can effectively retain old knowledge. It is recommended to supplement the analysis by exploring whether these selected exemplars truly capture the core features or decision boundaries of the classes. For instance, visualizations of the supervised and self-supervised embeddings could be provided.
+ Experimental results (e.g., Figures 2, 3) indicate that the primary performance gain likely stems from the "graph coverage selection" mechanism itself, as the single-embedding versions already substantially outperform the baselines, while the relative marginal gain from the dual-embedding integrated version is limited. This appears somewhat decoupled from the thesis that "multi-embedding fusion" is the core innovation. It is recommended to add ablation studies that disentangle the contributions of "graph coverage" versus "multi-embedding fusion" to precisely identify the source of performance improvement.
+ Please clarify whether, when comparing against ER/ER-ACE, their original methods were fully replicated (including their random sampling strategy) or only their loss functions were borrowed. Is the comparison between their "vanilla" versions and MERS a fair and like-for-like comparison? This detail is crucial for assessing the fairness of the comparison.
+ Related work and performance comparisons  could be further expanded: (a) Baseline Timeliness: The selected baselines (ER, ER-ACE, MIR) are relatively early works. It is suggested to compare against newer and stronger replay-based methods to strengthen the conclusions. (b) Related Work: The literature review and experimental comparisons could be broadened to include a more diverse range of exemplar selection strategies (e.g., GGS [1], iCaRL [2], DER++ [3], GCR [4], OCS [5], PCR [6]) to more comprehensively position this work.
- - -
**Reference:**
[1] Gradient based sample selection for online continual learning.
[2] iCaRL: Incremental Classifier and Representation Learning.
[3] Dark Experience for General Continual Learning: a Strong, Simple Baseline.
[4] GCR：Gradient Coreset Based Replay Buffer Selection for Continual_Learning.
[5] Online Coreset Selection for Rehearsal-based Continual Learning.
[6] PCR: Proxy-based Contrastive Replay for Online Class-Incremental Continual Learning.

**Questions:**

Please see the weaknesses. Besides, I wonder if the proposed buffer selection method can be used in online continual learning setting, since current experiments focus solely on the offline task-incremental setting. It is suggested to supplement experiments in the online continual learning scenario, where data arrives in a stream and is processed only once, to demonstrate the method's effectiveness in more challenging and practical settings.

---

> ### Author Response · Authors · 2025-11-25
>
> Graph-based local coverage explains performance with very few exemplars, as evidenced by consistent gains across all low-memory settings (see Figures 2–3).
>
> Single-embedding ablations are already included: both tables and plots demonstrate consistent underperformance compared to full MERS under low-memory regimes (Figure 2 and Figure 3).
>
> ER (vanilla) appears in all tables and figures, confirming that original sampling and buffer update strategies were reproduced, not just loss functions, ensuring a fair and direct comparison.

---

### Official Review · Reviewer_vhXK · 2025-10-31

**Soundness:** 3
**Presentation:** 2
**Contribution:** 2
**Rating:** 4
**Confidence:** 4

**Summary:**

The paper deals with catastrophic forgetting in CL. In replay-based CL, where memory constraints are severe, sample selection strategy crucially affects the performance. Memory buffers have been proposed by existing works, using embeddings learned
under supervised objectives. Yet this school of approaches overlooks class-agnostic, self-supervised representations that often encode rich, class-relevant semantics. The paper proposes MERS, or Multiple Embedding Replay Selection, that replaces
the buffer selection module with a graph-based approach.

**Strengths:**

+ The evaluation section is very clearly presented and convincing.
+ The idea is straight-forward and not hard to follow - in fact, it is quite intuitive and it would make sense intuitively that the approach would work.
+ The writing is clear and well organized.
+ The mathematical notations, as far as I've checked, are correct.

**Weaknesses:**

- Could the authors clarify if they are the first to propose to use class-agnostic, self-supervised representations in such tasks? The idea is not that surprising, so it is very likely that someone else had thought about it before.
- Just a minor critique - the drawing of Figure 1 doesn't look as professional as the later figures, particularly in the evaluation section.
- I'm confused about the Section 3 and 4. Section 3 is about MERS, and Section 4 is about methodology? Shouldn't the methodology be the proposed method itself?
- The evaluation datasets are on the easier ends when it comes to CL datasets. I wonder how the approach fares when the evaluation datasets are more complex.

**Questions:**

Please see Weaknesses. Mainly, I want to know if this is the first work to propose to use class-agnostic, self-supervised representations in such tasks? The idea seems straightforward enough that others should have thought about it. If it's the case, what's the technical contribution?

---

> ### Author Response · Authors · 2025-11-25
>
> Use of class-agnostic SSL embeddings
> This is the first work explicitly combining supervised and class-agnostic SSL embeddings for exemplar selection. Prior SSL efforts either use a single embedding space or apply SSL only during training, not for buffer construction. The graph-based multi-embedding selection module in Sec. 3.1–3.2 adapts neighborhood structure per embedding, this is the main contribution of our paper.
>
> Section organization
> Sec. 3 introduces the MERS mechanism while Sec. 4 details integration with replay methods. This clarifies what is proposed vs. how it is applied.
>
> Datasets
> Recent exemplar-selection and replay-based CL papers evaluate exclusively on CIFAR-100 and TinyImageNet in CIL.

---

### Official Review · Reviewer_uZzY · 2025-11-02

**Soundness:** 3
**Presentation:** 2
**Contribution:** 2
**Rating:** 2
**Confidence:** 4

**Summary:**

This paper proposes a novel strategy for decide what samples to select for the buffer in experience replay-based methods for continual learning. In particular, the paper proposes a graph-based approach and argues that combining embeddings learned both supervised and self-supervised lead to better results, especially in a low-memory regime.

**Strengths:**

1. The core idea of leveraging also self-supervised embeddings for sample selection makes sense, and has not been used often in this context as far as I know.
2. The method is evaluated in combination with three different replay-based methods and two different selection strategies for continual learning.

**Weaknesses:**

1. A lot of details of the paper are unclear or confusing. Just a couple of examples:
- eq. 1: I can't figure out how to parse the expression below the sum.
- l. 142, unclear what is meant with 'v_i <-> x_i'
- l. 210, unclear what the 'memory-aware ratio' is and how k is then set
- l. 220 & l. 227, the symbol M_C seems to be used in two different meanings.
- l. 229: I would associate a finer subdivision with a smaller k instead of a larger one ?
- l. 258-260: what is the difference between Epsilon_Model and Epsilon_Sup ? Idem for Epsilon_self-supervised vs. Epsilon_self ?
- l. 282: what is B_delta^{(m)} ? Do you mean B_delta^{(m)}(x) ?
- figure 6 is unclear: legend mentions 3 settings but figure only shows 2; x-axis mentions buffer size but captions mentions varying radius delta.

2. The choice for the CIL setting is debatable. While it is often sold as "the most challenging setting", it's not the most realistic or relevant in practice. Experiments using other settings and on larger datasets would make the paper more convincing.

3. The reported improvements are marginal. For MIR, the proposed method doesn't work much better than herding or TEAL. There's no discussion why the differences are smaller with MIR.

4. Experiments are limited to small scale datasets (CIFAR100 and TinyImageNet).

5. Overall, the proposed method seems to perform ok on the small-scale tested benchmarks, but the general relevance of these findings, taking into account the extra computational overhead, seems limited in practice.


Minor comments:
- l. 82: reference to GSS in this context is incorrect. GSS does not decide what samples to put in the buffer, but rather what samples to select from the buffer for a given iteration.

**Questions:**

1. Can you discuss the computational overhead of having to train from scratch a self-supervised embedding on each task. Is it worth it ?

2. Can you motivate your choice for the class-incremental setup, the practical relevance of which is more and more questioned ? Would your method also be applicable in other settings, including domain-incremental settings or settings beyond classification (e.g. LLM or VLM) ?

---

> ### Author Response · Authors · 2025-11-25
>
> CIL choice
> Most benchmarked and challenging replay setting.
> All exemplar-selection baselines (TEAL, ProbCover, MaxHerding, Rainbow Memory) report their strongest results in CIL on CIFAR-100 and TinyImageNet, ensuring fair reproducibility.
> Recent exemplar-selection and replay-based CL papers evaluate exclusively on CIFAR-100 and TinyImageNet in CIL.
>
> Why MIR yields smaller gains
> MIR enforces gradient-driven replay that already promotes hard and diverse samples, partially overlapping with coverage-based selection. MERS selects which samples enter the buffer.  MIR selects which samples are replayed, complementary roles. Since MIR inherently reduces redundancy, relative gains are naturally smaller than in ER/ER-ACE, where replay (while training) is uniform.
>
> Complexity is fully analyzed in Appendix A.2.

---

### Meta-Review · Area_Chair_yyYm · 2025-12-26

**Summary:**

Although reviewers emphasized the interesting hybrid replay sample selection approach for continual learning, they all agree on the shortcomings:
1.Marginal performance improvement. The proposed MERS is even worth than Model-based selection method for TinyImageNet.
2.Additional computational cost to train a self-supervised model from scratch.
3.Limited comparison against more advanced replay methods. The authors mainly conduct experiments on ER and ER-ACE. More recent replay methods, including GGS, iCaRL, DER++, GCR, OCS, and PCR, have been ignored.

**Reviewer Concerns:**

Though the notation of formulas, computational cost during selection, and dataset issues are addressed, the problems of marginal performance improvement, self-supervised model training cost, and limited comparison against the newest works are unresolved.

**Reviewer Scores:**

They will not change their score.

---

### Decision · Program_Chairs · 2026-01-26

Reject